# Structures of the *Escherichia coli* type 1 pilus during pilus rod assembly and after assembly termination

Paul Bachmann ●[1], Pavel Afanasyev ●[2], Daniel Boehringer ●[2] & Rudi Glockshuber ●[1] ✉

Uropathogenic *Escherichia coli* strains use filamentous type 1 pili to adhere to and invade uroepithelial cells. The pilus consists of a flexible tip fibrillum, formed by the adhesin FimH and the subunits FimG and FimF. The pilus rod is a helical assembly of up to 3000 copies of the main subunit FimA, terminated by a single copy of the subunit FimI that anchors the rod to the assembly platform FimD in the outer membrane. Although type 1 pilus assembly can be completely reconstituted in vitro, the precise mechanism of assembly termination on FimD is still unknown. Here, we present cryo-electron microscopy structures of the fully assembled pilus with all its components prior to and after incorporation of FimI, capped with the assembly chaperone FimC. The structures reveal that FimD positions the proximal end of the pilus rod at an angle of ca. 50 degrees relative to the plane of the outer membrane. Specific interactions between FimI and FimC, absent in the equivalent FimA-FimC interface of the non-terminated pilus, stabilize the assembly-terminated state. In addition, we present structures of the transition region between the tip fibrillum and the helical rod, showing how FimF aligns the tip fibrillum along the rod axis.

Chaperone-usher (CU) pili, named for the chaperone and usher (assembly platform) proteins that catalyze their assembly, are hair-like appendages displayed on the surface of numerous Gram-negative bacteria[1–3]. While type 1 and P pili of uropathogenic *Escherichia coli* are crucial virulence factors during urinary tract infections (UTIs)[4–6], other CU pili such as the archaic Csu pili of the multidrug-resistant pathogen *Acinetobacter baumannii* are required for biofilm formation[7,8]. Although the architectures of CU pili vary[9–12], they share a common assembly mechanism characterized by three key steps: i) to become assembly competent, newly synthesized pilus subunits entering the periplasm form a complex with a periplasmic chaperone (FimC for type 1 pili[13,14]) that accelerates their folding and prevents premature aggregation ii) pilus polymerization and secretion through consecutive addition of folded subunits is catalyzed by an assembly platform ("usher") located in the outer membrane (FimD for type 1 pili[15]), and iii) before polymerization can occur, assembly must be initiated by

incorporation of the most distal (pilus tip) subunit into the translocation pore of the usher[16]. CU pili carry tip adhesins for recognition of specific host cells, facilitating adhesion and colonization of host tissues, which is critical for establishing and maintaining infections[1]. The structural characterization of CU pili provides the basis for developing antibacterial drugs that either prevent pilus assembly or receptor binding[17].

The *E. coli* type 1 pilus is composed of a linear tip fibrillum at its distal end, formed by the mannose-binding adhesin FimH and the subunits FimG and FimF, and a helical pilus rod, composed of hundreds to several thousand copies of the main subunit FimA and a single copy of the assembly terminator FimI at its proximal end, capped with the assembly chaperone FimC[18–20]. The assembly platform FimD in the outer membrane of uropathogenic *E. coli* is the catalyst of type 1 pilus assembly[15]. It is composed of a 24-stranded β-barrel membrane domain that forms the translocation pore, which is sealed by a plug

[1]Institute of Molecular Biology and Biophysics, ETH Zürich, Otto-Stern-Weg 5, Zürich 8093, Switzerland. [2]Cryo-EM Knowledge Hub (CEMK), ETH Zürich, Otto-Stern-Weg 3, Zürich 8093, Switzerland. ✉e-mail: rudi@mol.biol.ethz.ch

domain only when the usher is inactive[16]. The periplasmic N-terminal domain of FimD (NTD) recognizes incoming chaperone-subunit complexes, while the two C-terminal domains (CTDs, encompassing CTD1 and CTD2) stabilize the growing base of the pilus for incorporation of the next subunit (Fig. 1a, b)[16,21,22].

Type 1 pilus subunits are structural homologs that share an incomplete immunoglobulin-like fold lacking the C-terminal β-strand (pilin domain, Fig. 1b). Each subunit carries an N-terminal extension (Nte, also termed donor strand) that completes the β-sheet fold of the preceding subunit by donor strand complementation[10,23] in which specific side chains of the Nte occupy five hydrophobic pockets (P1–P5) in the acceptor subunit[24]. In periplasmic complexes between pilus subunits and the chaperone FimC, the chaperone transiently complements the fold of the subunit[13]. This interaction is later replaced by the Nte of the next subunit during FimD-catalyzed subunit assembly, a process known as donor strand exchange[15,25–27].

The catalytic cycle of pilus assembly at the secretion platform FimD begins with the binding of an incoming chaperone-subunit complex to the FimD NTD[28]. The FimC-subunit complex is then transferred to the growing base of the pilus, which is stabilized by the FimD CTDs. Finally, the incoming subunit is incorporated into the pilus, displacing the FimC that capped the previous last pilus subunit, and the NTD of FimD becomes free for binding the next FimC-subunit complex. While only the FimH-FimC complex can activate FimD by displacement of the plug domain in the translocation pore, the order of incorporation of all following subunits is kinetically controlled in that native subunit-subunit interactions are most rapidly formed[20,29]. Once formed, subunit-subunit interactions are infinitely stable against dissociation under physiological conditions[30,31]. Pilus assembly terminates with the stochastic incorporation of the assembly terminator subunit FimI, which prevents the incorporation of further subunits and remains bound to FimC at the base of the pilus[20]. Since the translocation pore in FimD is only wide enough (-36–44 Å diameter[16]) to

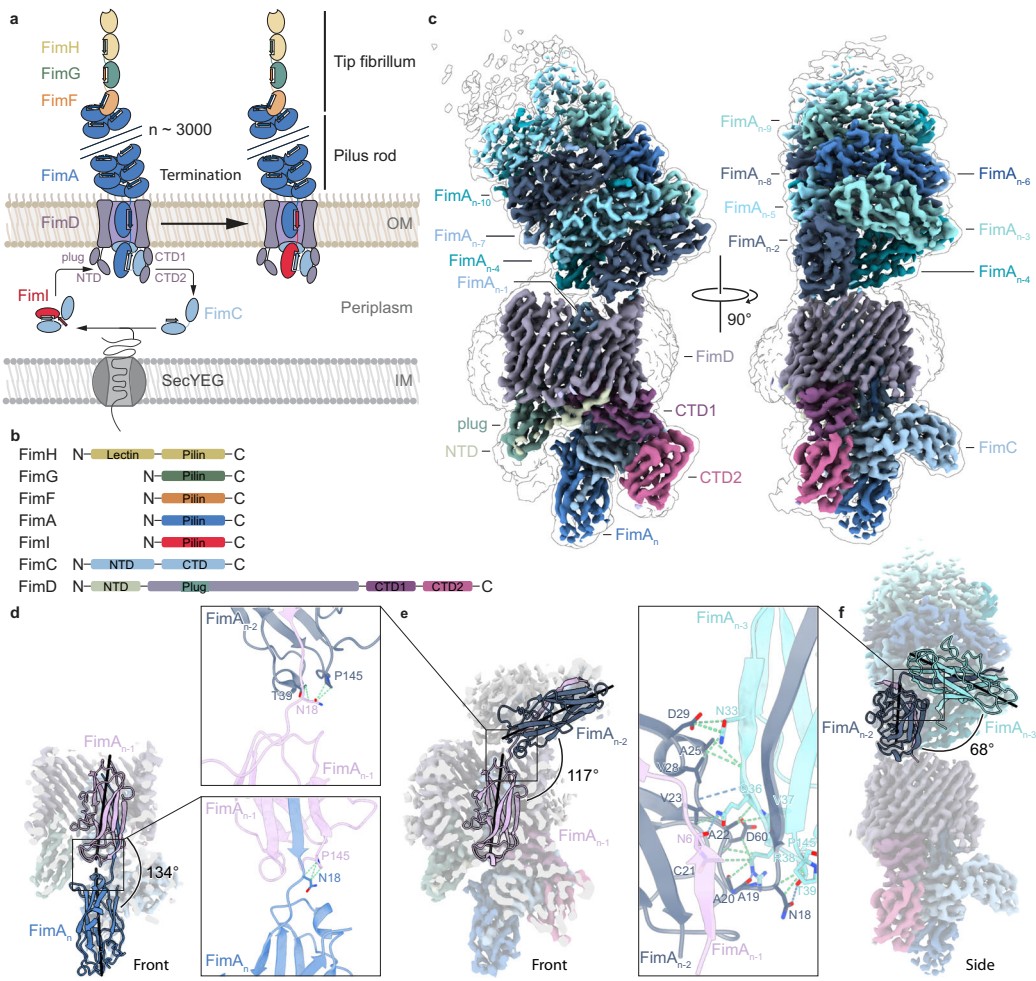

**Fig. 1 | Structure of the type 1 pilus during FimD-catalyzed assembly of FimA.** **a** Schematic overview of type 1 pilus biogenesis and assembly termination by FimI. All pilus subunits are transported through the inner membrane by the SecYEG translocon and fold in the periplasm once bound to the chaperone FimC. Only folded, FimC-bound subunits are assembly competent. FimI is incorporated as the last pilus subunit and terminates pilus assembly. **b** Schematic representation of the domain organization of all type 1 pilus components. **c** Cryo-EM map (composite of map II (pilus rod) and map III (assembly platform)) of the FimA-bound usher FimD in the process of pilus assembly. Cryo-EM density is colored based on the respective models II and III. **d** Angle and interface (inset) between subunit FimA$_n$ and FimA$_{n-1}$ in model IV. FimA$_n$ (navy blue) and FimA$_{n-1}$ (pink) are both shown as cartoon representations and map IV is shown in the background. Central axes of the two subunits

are shown as black lines. **e** Angle and interface (inset) between subunit FimA$_{n-1}$ and FimA$_{n-2}$ in model V. FimA$_{n-1}$ (pink) and FimA$_{n-2}$ (dark blue) are both shown as cartoon representations and map V is shown in the background. Central axes of the two subunits are shown as black lines. The insets in (**d**) and (**e**) show that there are only few potential hydrogen bonds stabilizing the FimA$_n$-FimA$_{n-1}$ and FimA$_{n-1}$-FimA$_{n-2}$ interfaces. The limited resolution in this area does not allow precise modeling of side chain conformations. **f** Angle and interface between subunit FimA$_{n-2}$ and FimA$_{n-3}$ in model II. FimA$_{n-2}$ (dark blue) and FimA$_{n-3}$ (blue green) are shown as cartoon representations and map II and III are shown in the background. Central axes of the two subunits are shown as black lines. The inset shows the hydrogen bonds that can be potentially formed by residues at the interface between FimA$_{n-2}$ (dark blue), the Nte of FimA$_{n-1}$ (pink) and FimA$_{n-3}$ (blue green).

accommodate the passage of a single folded subunit, the right-handed helical quaternary structure of the pilus rod, with a width of 72 Å (3.13 subunits/turn, 7.8 Å axial rise/subunit[10,32]) can only form on the extracellular side of the FimD translocation pore, thereby contributing to unidirectional subunit translocation.

Previously solved structures had captured states of FimD-catalyzed pilus assembly, shedding light on specific steps in the process. It was shown that all incoming subunit-chaperone complexes are initially bound by the NTD of FimD[28], that the first step of pilus assembly involves activation of the usher by the FimHC complex[15], and that incorporation of FimH triggers displacement of the plug domain to the periplasm by the FimH lectin domain, accompanied by a conformational change of the usher's transmembrane domain[16]. In the X-ray structure of the isolated FimDHC complex, FimHC interacts with the CTDs of the usher while the NTD is free to bind the next subunit[16]. A similar arrangement was observed in the crystal structure of the FimDHGFC complex[21]. Cryo-EM structures of the same complex revealed an additional conformation, where both the NTD and CTDs of FimD bind the FimFC complex after donor strand exchange has occurred[22]. These findings suggested a handover mechanism in which FimC-bound subunits are transferred from the NTD to the CTDs of FimD. Despite these insights into the mechanism of subunit incorporation catalyzed by the usher, critical aspects of type 1 pilus assembly remained unresolved. Specifically, there is no structural information on how the tip fibrillum connects to the pilus rod, how the major subunit FimA is incorporated into the rod via FimD, or how assembly is terminated with the incorporation of the final subunit, FimI.

In this work, we provide a comprehensive structural description of the full mechanism of type 1 pilus assembly and termination. Using cryo-EM, we determine structures of the assembly platform FimD in the process of pilus rod assembly, the transition of the tip fibrillum to the pilus rod, and the FimI-terminated state of FimD. These findings offer a detailed view of the complete assembly process of the pilus, advancing our understanding of this essential UTI virulence factor.

## Results

### FimA incorporation into growing pili

Our primary objective was to solve the structure of the entire type 1 pilus complex and to elucidate the mechanism underlying its assembly. The key step in pilus rod formation is the export of the major pilin subunit FimA by the usher FimD. To investigate this step, we made use of an established protocol to assemble type 1 pili in vitro from purified components (Supplementary Fig. 1)[20]. The assembly conditions were further optimized to produce a sample with an average pilus length suitable for cryo-EM, allowing us to determine the overall structure of the FimDHGFA$_n$C complex to 4.1 Å resolution (Fig. 1, Supplementary Figs. 2, 3a, b, Supplementary Table 1).

The 3D-reconstruction of the complex (map I) revealed that the pilus rod sits at an angle on top of the usher. However, the rod was initially not well-resolved due to its flexible connection between subunit FimA$_{n-1}$ occupying the pore and FimA$_{n-2}$ emerging from the pore. To address this, multiple data processing strategies in single-particle analysis were employed, which significantly improved the resolution of both the pilus rod and the FimA-bound usher (Supplementary Fig. 2). Multi-body refinement in RELION-5.0[33,34] further enhanced the resolutions, yielding 3.6 Å for the pilus rod (body 1) and 3.9 Å for the assembly platform (body 2) (Supplementary Figs. 2, 3c–f, 4b, c, 5, Supplementary Table 1). By combining the maps of the two bodies, we constructed a composite map that includes both the assembly platform and pilus rod, which sits at an approximately 51° angle on top of the usher (Fig. 1c, Supplementary Figs. 4a, 5c).

Multi-body refinement provided insights into the movement of the pilus rod relative to the assembly platform. Although restrained by the rim of the β-barrel and the extracellular loops of FimD, principal

component analysis revealed a twisting motion of up to 11°, as well as side-to-side and back-and-forth motions of up to 19° and 18°, respectively (Supplementary Fig. 5a–f). This flexible connection likely enhances the range of the pilus, enabling it to bind mannose binding sites on the epithelial host cell surface more effectively. Furthermore, it would also allow for simultaneous binding of multiple pilus rods of varying lengths to the same surface, promoting efficient host colonization.

Next, we analyzed the process of FimA subunit incorporation into the growing pilus rod. Using a local refinement approach, we obtained a 3D-reconstruction at 3.5 Å resolution, capturing the usher along with FimC and the last two incorporated subunits, FimA$_n$ and FimA$_{n-1}$ (Fig. 1d, Supplementary Figs. 3g,h, 4d, Supplementary Table 1). Additionally, we generated another 3D-reconstruction at 3.6 Å resolution, which also included FimA$_{n-2}$ (Fig. 1e, Supplementary Figs. 3i,j, 4e, 6a, Supplementary Table 1). On the periplasmic side of FimD, the FimA$_n$-FimC complex is bound to the CTDs, while the preceding subunit, FimA$_{n-1}$, occupies the pore of the usher. The NTD is unoccupied and not well-resolved.

The overall arrangement of the subunits FimA$_n$, FimA$_{n-1}$ and FimC in the FimDHGFA$_n$C complex is similar to that observed for conformer 2 of the FimD-tip complex FimDHGFC (PDB-ID 6E15[9]), but subtle upward displacements relative to the transmembrane domain of FimD are observed for several subunits. Specifically, subunit FimA$_n$ is positioned about 4.5 Å higher than FimF, FimC is moved upwards by roughly 8 Å, and CTD2 of FimD is shifted upwards by approximately 4 Å (Supplementary Fig. 6d). This previously unknown FimD conformation appears to be specific to FimD-rod complexes, in which another notable change occurs in the orientation of the Nte of the FimA$_{n-2}$ subunit emerging from the usher pore. While no major conformational changes between the pilin domains of FimA$_{n-1}$ and FimA$_{n-2}$ can be observed (Supplementary Fig. 6c), the relative angle of the Nte to the succeeding FimA subunit changes dramatically (Fig. 1e). This adjustment appears to still be constrained by the upper rim of the usher, preventing FimA from adopting its final orientation found in the pilus rod (Supplementary Fig. 6e). Multi-body analysis indicates that the connection between FimA$_{n-1}$ and FimA$_{n-2}$ serves as the main pivot point for pilus rod motion relative to the usher (Supplementary Fig. 5). Importantly, it is only between FimA$_{n-2}$ and FimA$_{n-3}$ that the characteristic angle observed between adjacent FimA subunits within the mature pilus rod can finally be adopted (Fig. 1f).

As a consequence of the fact that incorporation of the first FimA subunit is rate limiting for pilus rod assembly[20], we obtained a mixture of FimD-rod complexes and FimDHGFC complexes that had not yet incorporated a single FimA after completion of FimD-catalyzed tip assembly.

While the previously reported handover conformation of the FimDHGFC complex (PDB ID 6E14[22]) was not observed in rod-containing structures within this mixture, our analysis revealed two analogous conformations of FimDHGFC complexes (Supplementary Fig. 7, Supplementary Table 1; FimDHGFC conformer 1: Supplementary Fig. 8a–d, FimDHGFC conformer 2: Supplementary Fig. 8e–h). In conformer 1, the structure closely resembles PDB-ID 6E15[22], with the NTD remaining unoccupied. In contrast, both the NTD and the CTDs are engaged in binding of the last chaperone-subunit complex in conformer 2. Conformer 2 differed from the previously reported handover state in PDB-ID 6E14[22] and is more similar to an intermediate secretion step observed for the tip adhesin FimH (PDB-ID 9BOG[35]). However, the resolution did not allow unambiguous assignment of the incorporated subunits (Supplementary Fig. 8h).

The FimD-rod and FimDHGFC structures provide further insight into pilus assembly and suggest that FimD quickly adopts the post-handover conformation on the periplasmic side, driven by quaternary structure formation of the pilus rod on the extracellular side. Furthermore, the observation that the growing pilus rod is oriented at a

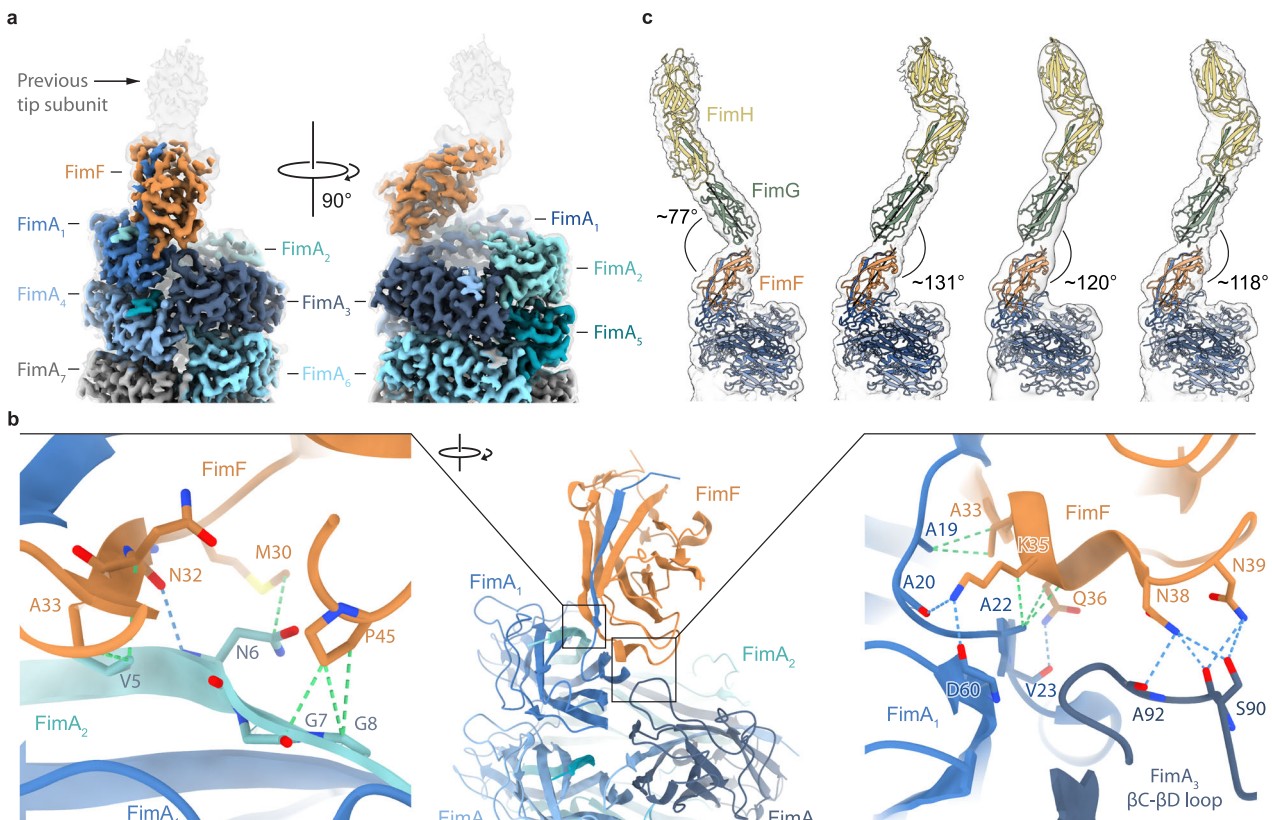

**Fig. 2 | The role of FimF in aligning the tip fibrillum with the rod. a** Cryo-EM map of the transition from the helical FimA polymer (rod) to the last subunit of the tip fibrillum, FimF (map VIII). The unsharpened map is shown in the background at low threshold level to show the positioning of the previously incorporated subunit FimG. FimF is colored orange, while the first seven FimA subunits (FimA₁-FimA₇) are colored in shades of blue. FimF is sitting on top of the pilus rod and is tilted upwards while making contacts with FimA₁, FimA₂, and FimA₃. **b** Cartoon representation of FimF-FimA₁-FimA₂-FimA₃-FimA₄ (center, model VIII). The left inset shows potential hydrogen bonds (blue dashed lines) and hydrophobic contacts (dashed green lines) between FimF (orange) and the donor strand of FimA₂ (blue green) which

complements the fold of FimA₁ (middle blue). The right inset shows interactions between FimF and both FimA₁ and FimA₃ (dark blue) (see Supplementary Fig. 9 for densities). **c** Conformational variability of the tip fibrillum, with focus on the angle between FimF and FimG. Four different refined cryo-EM maps from focused 3D classification are shown at the same orientation. The cryo-EM model from a) was rigid-body fitted in each map as well as either an Alphafold2 model of the FimH-FimG-FimF complex (Conformer 1, map IX) or FimH and FimG of 3JWN (Conformer 2–4, map X-XII). The central axis of each FimF and FimG subunit is shown as a black line. Angles between FimF and FimG subunits are indicated.

distinct angular range during assembly ties in well with the recent finding that FimD is required for formation of the most stable pilus rod quaternary structure[31]. Our findings suggest that the overall architecture of the usher guides the formation of the correct rod structure by orienting the rod such that exiting FimA subunits directly slot in place. Previous studies suggested that subunits follow a low-energy pathway through the usher's translocation pore, undergoing a twisting motion as they are translated through the pore[21]. The pilus rod is indeed positioned relative to FimD such that the subunit emerging from the pore (n-1 to n-2 position) can form contacts with the subunit transitioning from position n-4 to n-5, which represents the main stacking interface between the pilus rod subunits (Supplementary Movie 1)[10].

## FimF and the tip-to-rod transition

A previous study solved the crystal structure of a recombinantly produced, soluble tip fibrillum comprising FimH, FimG and two FimF subunits, with the last FimF complemented by FimC[23]. In addition, the cryo-EM structure of the FimDHGFC complex uncovered the tip fibrillum bound to the assembly platform[22], and NMR studies suggested a dynamic subunit-subunit interface in FimF-FimF and FimG-FimF complexes[36]. Here, we focused on determining the missing structure of the tip-to-rod transition and analyzing its influence on the flexibility of the tip fibrillum.

We determined the cryo-EM structure of the transition from the last tip fibrillum subunit, FimF, to the first turn of the pilus rod at 3.1 Å resolution (Fig. 2a, Supplementary Fig. 9a–d, Supplementary Table 1). The structure revealed key interactions between FimF and the first three FimA subunits (FimA₁, FimA₂, and FimA₃), which are well-resolved and clearly identifiable. In contrast, the preceding subunit of the tip fibrillum, FimG, appeared flexible and was only visible at lower thresholds, highlighting its dynamic connection to FimF (Fig. 2a).

Specifically, predominantly hydrophobic interactions were found between FimF (complemented with the donor strand of FimA₁) and the donor strand of FimA₂ (Fig. 2b, left panel), while the interfaces of FimF with FimA₁ and FimA₃ included both hydrophobic contacts and hydrogen bonding (Fig. 2b, right panel). Notably, superimposition of the first three FimA subunits revealed a conformational shift in the βC-βD loop: while solvent-exposed in FimA₁ and FimA₂, the loop in FimA₃ adopts a different conformation that enables interaction with FimF through five additional hydrogen bonds (Fig. 2b, right panel, Supplementary Fig. 10a). These interactions are stabilized by hydrogen bonds between Lys35 and Gln36 of FimF with backbone carbonyl oxygens of specific residues in FimA₁, including Ala20, Asp60 and Val23. Importantly, the chemical identity of Lys35 and Gln36 is conserved across FimF homologs of other chaperone-usher pili at the transition from the tip fibrillum to the rod (Supplementary Fig. 10b).

Further comparison of the FimF-FimA$_I$ interface with the interface of two adjacent FimA subunits within the pilus rod (FimA$_{n-4}$ and FimA$_{n-5}$) revealed a similar hydrogen-bonding mode. In both cases, a basic residue (Arg38 of FimA$_{n-5}$) interacts with backbone carbonyl groups (Ala19 and Ala20 of FimA$_{n-4}$) (Supplementary Fig. 10c, d). However, Arg38 and Gln36 are shifted by a few residues when aligned to sequences of other FimF homologs, consistent with the substantially different orientation of adjacent FimA rod subunits compared to the FimA-FimF interface. Thus, these differences in the interfaces and the simultaneous interaction of FimF with FimA$_1$, FimA$_2$ and FimA$_3$ are likely incompatible with a FimA-like orientation within the helical rod and thereby facilitate the transition between the linear tip fibrillum and the helical rod architecture.

To further investigate whether the sequence conservation of FimF homologs has structural implications on tip fibrillum-to-rod transitions, we performed Alphafold2[37] predictions of all transitions included in the sequence alignment in Supplementary Fig. 10b. Models of the tip-to-rod transitions in F1C/S pili, F9 pili, P pili, and Yfc pili show that all FimF homologs point upward in a manner similar to FimF (Supplementary Fig. 11). This underlines the critical role of FimF homologs that function as adaptors aligning the tip fibrillum with the axis of the pilus rod. The extended conformation of the FimF adaptor and the rest of the tip fibrillum may also function to increase the accessibility of FimH to buried high-mannose type N-glycans on host cells.

Moreover, FimF and its homologs may also play a role in facilitating the formation of the first turn of the pilus rod. A model of FimF, FimA$_1$, FimA$_2$ and FimA$_3$ positioned atop the usher FimD suggests that FimF could stabilize the first turn of the rod by making additional contacts to FimA$_3$ forming a template for the quaternary structure of the pilus rod (Supplementary Fig. 10d).

Finally, we analyzed the position of the subunit FimG relative to the well-resolved FimF subunit within the tip fibrillum. While FimG initially appeared flexible in the consensus refinement, further 3D classification without alignment in RELION-5.0 yielded distinct classes in which FimG and FimH could be fitted (Fig. 2c, Supplementary Fig. 9e, Supplementary Table 1). Interestingly, the angle relative to FimF varies between -77° and -131°. This flexibility is likely advantageous for accessing and binding to the slightly buried N-glycans of Uroplakin Ia[38]. Our findings concerning the flexibility between FimF and its preceding subunit agree with previous NMR studies that revealed dynamic interfaces between subunits of the tip fibrillum[36].

## Assembly termination by FimI

To explore the structural implications of FimI-mediated termination of pilus assembly, we performed a FimD-catalyzed assembly reaction of pilus rods in vitro similar to the one for the FimDHGFA$_n$C complex, except that excess FimIC$_{His}$ was also added at the end of the reaction (Supplementary Fig. 1). Using Ni-NTA affinity chromatography, we separate complexes with a terminal FimIC$_{His}$ subunit from complexes with terminal FimA$_n$C and determine the structure of the FimI-terminated assembly platform at 3.6 Å resolution (map XIII, Supplementary Figs. 12, 13a, b, Supplementary Table 2).

Further 3D classification was performed to also resolve the unresolved rod subunits on the extracellular side of the usher, revealing increased compositional heterogeneity compared to the FimDHGFA$_n$C sample (Supplementary Fig. 12). Since a ten-fold excess of FimIC$_{His}$ over FimD was used to ensure saturation with FimI, we also observed non-physiological states in 3D classification, such as complexes lacking pilus rods due to premature FimI incorporation after FimF, or complexes containing two terminal FimI subunits (FimDHGFA$_n$I$_2$C$_{His}$ complex, maps XVII and XVIII, Supplementary Fig. 12, Supplementary Figs. 13i–l, 15a–c and Supplementary Table 2). Given the low FimI concentrations in the periplasm (FimA:FimI ratio ≥100:1)[20], we focused our analysis on physiological pilus complexes with a single terminal FimI subunit capped by FimC$_{His}$. A 3D

reconstruction of the FimDHGFA$_n$IC$_{His}$ complex at 6.4 Å resolution was generated (Map XIV, Supplementary Figs. 12, 13c, d, Supplementary Table 2). Initial reconstructions showed poor resolution of both the pilus rod and the assembly platform. However, multi-body refinement with blush regularization in RELION-5.0[33,34] improved the resolution to 4.0 Å for the pilus rod (map XV) and 4.3 Å for the FimI-bound assembly platform (map XVI) (Fig. 3a, Supplementary Fig. 13e–h, Supplementary Table 2). This refinement also enabled analysis of rod flexibility relative to the FimI-terminated assembly platform.

The resulting composite model (Fig. 3b) shows that the FimI-terminated pilus rod sits on top of the assembly platform at an angle of about 48°, similar to that observed for the FimDHGFA$_n$C complex (Supplementary Figs. 14b–d). Despite a greater degree of motion in the FimI-bound state, the observed modes of pilus rod motion and its conformation remained highly similar to those in the complex with a terminal FimA subunit (Supplementary Fig. 16).

In addition, the comparison of the complexes with and without FimI showed that the subunit in the usher pore and the FimC-complemented subunit on the periplasmic side occupy nearly the same positions when the two structures are aligned based on the transmembrane and plug domains of the usher (Fig. 3c, RMSD = 0.56 Å). The overall conformation of the usher also remains unchanged between the FimA and the FimI-bound states. In both models, the NTD is flexible and only visible at lower threshold, while both CTDs specifically interact with the terminal subunit-chaperone complex.

To test whether conformational rearrangements in FimD are associated with assembly termination, we compared FimD in the FimI-terminated complex (model XVI) with FimD in the complex lacking FimI (model IV) and FimD in complex with the tip fibrillum with FimF as the last subunit (PDB-ID 6E15[22]). In contrast to the major conformational changes observed in early steps of pilus assembly after plug domain displacement (Supplementary Fig. 17a, b)[16], no major conformational changes were detected in the transmembrane domain of FimD upon incorporation of FimI (Supplementary Fig. 17c–h).

We then aligned the FimDHGFA$_n$IC$_{His}$ complex (model XVI) to the previously crystallized FimAIC$_{His}$ complex (PDB-ID 6SWH[20]) via FimC to assess structural differences. The angle between FimA and FimI differed: in the isolated FimAIC$_{His}$ complex, which is not restrained by FimD, FimI adopts a slightly different angle that would cause a clash with the transmembrane domain of FimD (Supplementary Fig. 18). The interface between FimC and FimA in the FimAIC$_{His}$ complex was previously proposed to slow FimC dissociation, preventing further assembly and anchoring of the pilus to the outer membrane[20]. In our FimI-terminated pilus complex (model XVI), we observed a similar interface, however, this interface is also present in the FimA-bound complex (model IV) without major differences (interface areas: 266 Å$^2$ for model XVI, 263 Å$^2$ for model IV) (Fig. 3d).

To identify structural parameters explaining assembly termination by FimI, we next analyzed the interfaces between FimC and FimA (model IV), FimC and FimI (model XIII), and FimC and FimF (PDB-ID 6E14[22]) in the respective FimD complexes (Fig. 4a–c). While the interactions between FimC and the respective subunit (FimA, FimI, FimF) are primarily mediated through backbone hydrogen bonding involving the G1 strand of FimC, differences emerge in the interactions with FimA strand A′. In contrast to FimA and FimF, FimI does not only form hydrogen bonds with the neighboring A1′ strand of FimC, but also with the B1 strand of FimC. Here, Arg25 of FimI additionally interacts with Thr23 and Asn25 of the B1 strand of FimC via hydrogen bonds (Fig. 4 - insets). In general, the total interface area between FimI and FimC (1739 Å$^2$) is larger than that of FimA and FimC (1562 Å$^2$) or FimF and FimC (1449 Å$^2$) and involves more potential hydrogen bonds (FimI-FimC: 30 H-bonds, FimA-FimC: 21 H-bonds, FimF-FimC: 19 H-bonds).

Previous work on PapH, the terminating subunit of the related P pili from uropathogenic *E. coli*, showed that donor strands from P pilus subunits could not displace the pilus chaperone PapC from PapH in

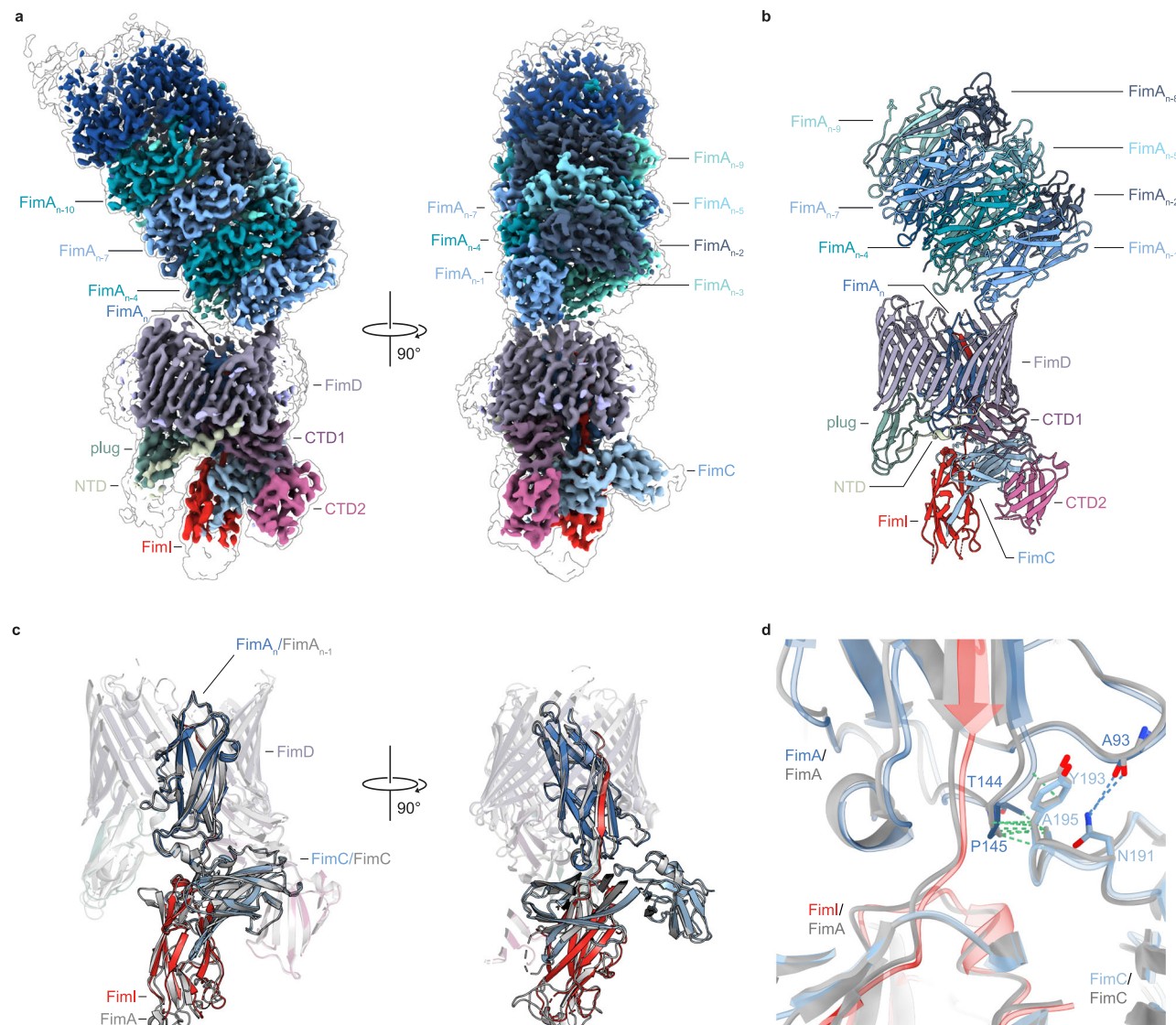

**Fig. 3 | Structural basis of type 1 pilus assembly termination through incorporation of FimI. a** Composite cryo-EM map of the FimDHGFA$_n$IC$_{His}$ complex (map XV (pilus rod) and map XVI (assembly platform)). **b** Cartoon representation of the FimDHGFA$_n$IC$_{His}$ complex (composite of model XV and model XVI). The complex is colored according to the color scheme displayed in **a**. **c** Structural alignment of the FimA-bound (FimDHGFA$_n$C complex, model IV) and FimI-bound usher (FimDHGFA$_n$IC$_{His}$ complex, model XVI) based on the transmembrane and plug domain of FimD (residues 140–665; RMSD = 0.56 Å). The FimA-bound usher is colored gray while the FimI-bound usher is colored according to the same color scheme as in **a**. **d** Detail of the alignment in (**c**) displaying the interaction between the FimA subunit occupying the pore and the chaperone FimC in the FimA-bound (gray) and FimI-bound (colored) usher complexes. The patterns of potential inter-subunit hydrogen bonds are nearly identical in both structures.

vitro[39]. This was attributed to the absence of a defined P5 pocket in the X-ray structure of the PapD-PapH complex, the presumed attacking point of the donor strand[40]. A closed P5 pocket was also observed in the crystal structure of the FimAIC$_{His}$ complex[20]. Indeed, we also identified a closed P5 pocket in the FimD-bound FimIC complex. However, when comparing the usher complexes with FimA or FimI as the last incorporated subunit with the usher-tip fibrillum complex (last subunit: FimF, with open P5 pocket), we found that the P5 pocket was closed in the terminal FimA subunit as well as in FimI (Supplementary Fig. 19)[22]. We conclude that a closed P5 pocket may reduce the efficiency of donor strand exchange with other subunits, but it cannot be the sole mechanism for assembly termination. In the case of the related Caf1 fibers (F1 capsule) from *Yersinia pestis*, for example, a closed P5 pocket in the subunit Caf1 complexed with the chaperone Caf1M still allows efficient assembly of Caf1 polymers[41]. It could also be that the acceptor cleft of FimI is simply not compatible with the donor strand of FimA in the terminated type 1 pilus, making termination permanent.

In any case, a much stronger binding of FimC to the terminal FimI subunit compared to a terminal FimA due to a larger interface and a higher number of inter-molecular hydrogen bonds appears to be critical for assembly termination by FimI. This view is also consistent with a previous study showing that FimC spontaneously dissociates 220-fold slower in solution from the ternary FimAIC$_{His}$ complex than from a binary FimIC$_{His}$ complex, and 13-fold slower compared to the FimAAC complex[20].

## Discussion

This study presents multiple structural insights into type 1 pilus assembly at previously uncharacterized stages of the assembly reaction and provides a nearly complete picture of the FimD-catalyzed pilus assembly process. Specifically, we show that FimD plays a crucial role in positioning FimA subunits emerging from the translocation pore such that i) the thermodynamically most stable interface between adjacent FimA subunits and the first turn of the helical rod structure

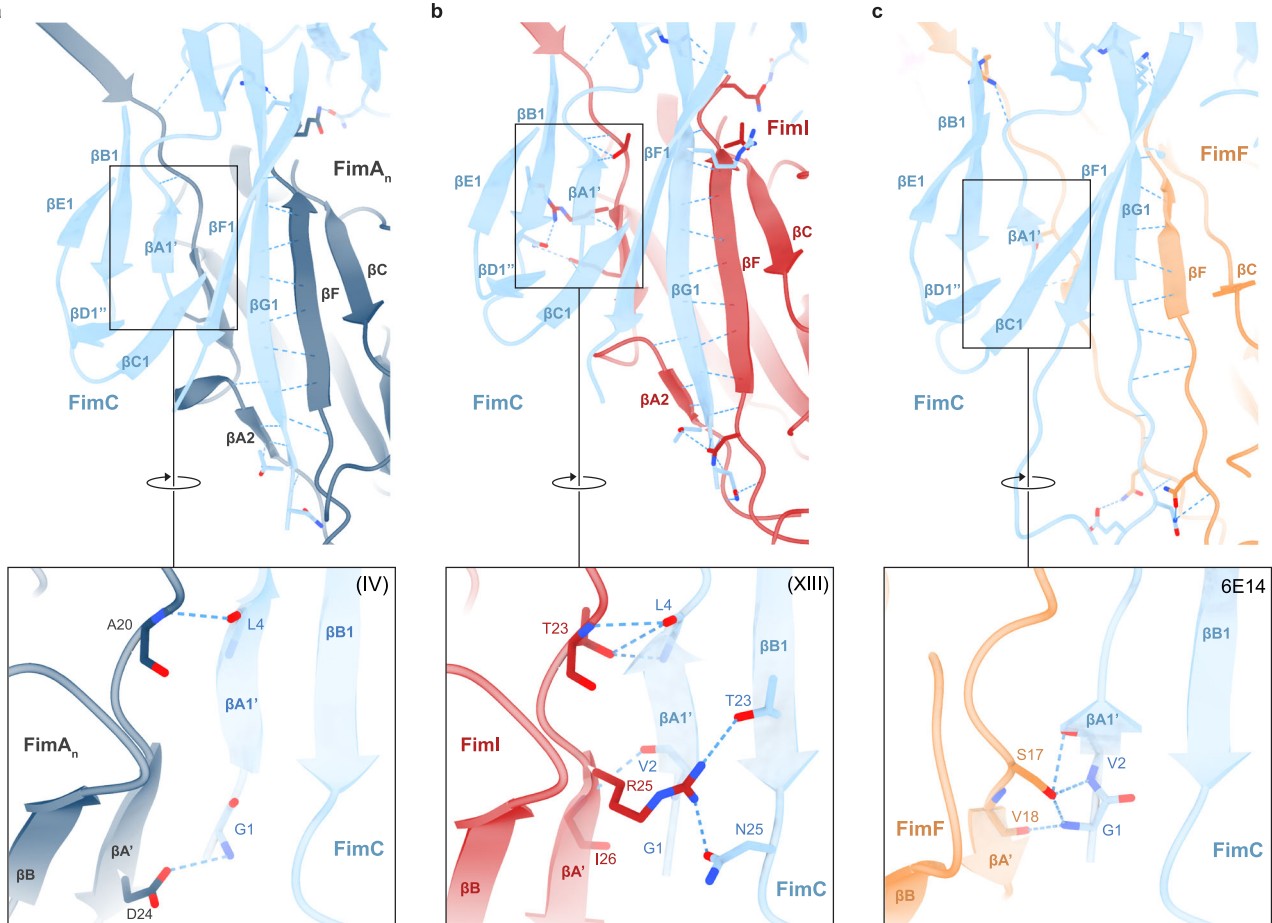

**Fig. 4 | Comparison of interfaces between the last incorporated subunit and the chaperone FimC in complexes FimDHGFA$_n$C, FimDHGFA$_n$IC$_{His}$ and FimDHGFC.** **a** Overview of interactions between subunit FimA$_n$ and FimC in the FimDHGFA$_n$C complex (model (IV)). Hydrogen bonds are displayed as dashed blue lines. Most of the hydrogen bonds are found between the G1 strand of FimC and the neighboring F and A2 strands of FimA$_n$. A second interface displayed in the inset can be found between residues of the A1' strand of FimC and the A' strand of FimA$_n$. **b** Overview of interactions between subunit FimI and FimC in the FimDHGFA$_n$IC$_{His}$ complex (model (XIII)). Hydrogen bonds are displayed as dashed blue lines. Again, most of the hydrogen bonds are found between the G1 strand of FimC and the neighboring F and A2 strands of FimI. A second interface (inset) can be found between residues of the A1' and B1 strand of FimC and the A' strand of FimI. **c** Overview of interactions between subunit FimF and FimC in the FimDHGFC complex (PDB-ID 6E14[22]). Hydrogen bonds are displayed as dashed blue lines. Again, most of the hydrogen bonds are found between the G1 strand of FimC and the neighboring F strand of FimF. A second interface (inset) can be found between residues of the A1' strand of FimC and the A' strand of FimF.

can only form when at least 5 FimA copies are incorporated into the growing pilus (Supplementary Movie 1) and ii) that the main orientation of the assembled pilus rod is at an angle of ca. 50° relative to the plane of the *E. coli* outer membrane. This angular orientation of the rod, as opposed to a perpendicular orientation, is predicted to strongly increase the search space of the tip fibrillum for mannosylated surface receptors on uroepithelial target cells assuming that the entire FimD-anchored pilus is mobile and can rotate within the membrane. For the same reason, the angular rod orientation might also modulate the strength of adhesion of piliated uropathogenic *E. coli* to target cells by e.g. dampening tensile mechanical forces that trigger the previously described catch bond formation between FimH[42–44] and its receptors in addition to force-induced pilus uncoiling[45,46].

Moreover, the solved structures of the transition from the distal end of the pilus rod to the tip fibrillum revealed a critical role of FimF in the assembly reaction. FimF not only orients the linear tip fibrillum along the rod axis but promotes formation of the first helical turn of the rod in the initial stage of rod assembly through interacting with the rod's first three FimA subunits, creating the template for elongation of the helical rod polymer. A critical role of FimF for FimD-catalyzed pilus rod formation is further supported by the fact that FimD cannot

catalyze FimA rod assembly when it bears FimH instead of FimF at the growing pilus end[20].

In the solved structures of FimD-bound pilus rod complexes, the FimA subunits n–n$_{-2}$ at the proximal end of the rod are essentially only held-together via donor strand complementation. This is consistent with previous studies showing that the interactions responsible for donor strand complementation between adjacent FimA subunits are more resistant to mechanical forces than the stacking interactions within the helical rod, which begin to unwind at comparably low mechanical forces[46–49].

Nevertheless, formation of the helical rod structure clearly drives unidirectional subunit translocation, analogous to the previously described self-secretion mechanism of archaic Csu pili in which the zig-zag architecture of the pili is frozen by a specific clinch mechanism[11]. Immediate incorporation of translocated FimA subunits into the helical rod structure may also explain why the handover-conformation of FimD detected previously as an assembly intermediate of tip fibrillum assembly[22] was not significantly populated in preparations of FimD complexes bearing pilus rods. Together, the presented structures of the completely assembled type 1 pilus and the finding that assembly termination by FimI is primarily caused by strong interactions between

FimC and FimI should provide a rational basis for developing pilus assembly inhibitors for UTI treatment.

## Methods

### Protein production and purification

**Purification of FimDCH.** Purification of FimDCH was performed as described previously[16,50]. In brief, *E. coli* Tuner cells transformed with plasmids pAN2-Strep encoding for FimD and pETS1001 encoding for FimH and FimC$_{His}$ were grown in TB medium supplemented with 30 µg/mL Kanamycin and 100 µg/mL Spectinomycin at 37 °C. Protein expression was induced by addition of IPTG to a final concentration of 100 µM and L-arabinose to 0.1% (w/v) at an $OD_{600}$ of 0.8–1.0. After addition of glycerol to 0.1 % (v/v), cells were grown for 48 h at 16 °C and then harvested by centrifugation (9180 × *g*, 10 min, 4 °C).

Cells were resuspended in 20 mM Tris-HCl [pH 8.0 at 4 °C] containing cOmplete protease inhibitor, EDTA-free tablets (Roche) using a homogenizer and disrupted using a microfluidizer. Cell debris was pelleted by centrifugation (10 min, 5000 × *g*, 4 °C) and N-lauroylsarcosine was added to the supernatant to a final concentration of 0.5% (w/v) to solubilize the inner membranes. After stirring for 5 min at room temperature (RT), the outer membrane (OM) was pelleted using ultracentrifugation (1 h, 100,000 × *g*, 4 °C) and then resuspended in 20 mM Tris-HCl [pH 8.5 at 4 °C], 120 mM NaCl and protease inhibitors (9 mL of buffer/g of OM pellet). The OM fraction was solubilized by addition of 1.5% n-dodecyl-β-D-maltopyranoside (DDM) and stirring overnight at 4 °C. Insoluble material was removed by ultracentrifugation (1 h, 100,000 × *g*, 4 °C) and the supernatant was loaded onto a 5 mL HisTrap HP column (Cytiva) equilibrated with 20 mM Tris-HCl [pH 8.5 at 4 °C], 120 mM NaCl, 0.05% (w/v) DDM (buffer A). The column was first washed with buffer A containing 25 mM imidazole and the protein was eluted with buffer A containing 250 mM imidazole. Eluted protein was diluted 2-fold in buffer A and loaded onto an 8 mL Strep-Tactin Sepharose gravity flow column (IBA Lifesciences GmbH) equilibrated with the same buffer. After washing with buffer A, FimDCH was eluted by addition of buffer A containing 2.5 mM D-desthiobiotin. Finally, the protein was concentrated using an Amicon Ultra filter with a molecular weight cutoff (MWCO) of 10 kDa (Merck Millipore) and applied to a Superdex 200 16/60 size-exclusion chromatography column (Cytiva) equilibrated with 20 mM Tris-HCl [pH 8.0 at room temperature], 50 mM NaCl, 0.05% DDM. Appropriate fractions containing FimDCH were pooled after size-exclusion chromatography, flash-frozen in liquid nitrogen and stored at −80 °C before being used in in vitro assembly experiments.

### Purification of FimFC and FimGC

Purification of FimFC and FimGC was performed as described previously[20]. In brief, for production of FimFC and FimGC, HM125 *E. coli* cells were transformed with plasmid pFimFC-ATG-trc or pFimGC-trc and grown at 30 °C in 2YT medium supplemented with 100 µg/mL Ampicillin. Protein production was induced at an $OD_{600}$ of 0.7 by addition of IPTG to a final concentration of 100 µM. Cells were grown for an additional 4 h and then harvested by centrifugation (9180 × *g*, 10 min, 4 °C). Cells were resuspended in 20 mM Tris-HCl [pH 7.5 at 4 °C], 150 mM NaCl, 5 mM EDTA and 1 mg/mL polymyxin B sulfate and incubated under shaking conditions at 4 °C for 1 h. The resulting periplasmic extract was cleared by centrifugation (48,000 × *g*, 30 min, 4 °C).

For FimFC, the supernatant was dialyzed against 20 mM Tris-HCl [pH 8.0 at 4 °C], centrifuged (48,000 × *g*, 10 min, 4 °C) and the supernatant was loaded onto a Q Sepharose FF column (CV = 15 mL; Cytiva) equilibrated with the same buffer. FimFC was collected in the flowthrough and solid ammonium sulfate was added to a concentration of 1.2 M. After pelleting insoluble material (48,000 × *g*, 10 min, 4 °C), FimFC was loaded onto a Phenyl Sepharose HP column (CV = 10 mL; Cytiva) equilibrated with 20 mM Tris-HCl [pH 8.0 at 4 °C], 1.2 M $(NH_4)_2SO_4$. A linear gradient from 1.2 M to 0 M $(NH_4)_2SO_4$ was used to elute FimFC. Pooled fractions containing FimFC were dialyzed against 20 mM MES-NaOH [pH 5.5 at 4 °C], loaded onto a Source 30S column (CV = 17 mL; Cytiva) equilibrated with the same buffer and eluted with a linear gradient from 0 to 200 mM NaCl. Fractions containing FimFC were pooled, concentrated using a 10 kDa MWCO Amicon Ultra filter (Merck Millipore) and applied to a Superdex 75 16/60 size-exclusion chromatography column (Cytiva) equilibrated with 20 mM Tris-HCl [pH 8.0 at room temperature], 50 mM NaCl. After size-exclusion chromatography, fractions containing FimFC were pooled, flash-frozen in liquid nitrogen and stored at −80 °C before being used in in vitro assembly experiments.

For FimGC, the cleared periplasmic extract was applied to the same Q Sepharose FF column but eluted using a linear gradient from zero to 400 mM NaCl. Fractions containing FimGC were pooled, dialyzed against 20 mM MOPS-NaOH [pH 6.7 at 4 °C] and applied to a Resource S column (CV = 6 mL; Cytiva) equilibrated with the same buffer. A linear gradient from 0 to 200 mM NaCl was used to elute the protein. Solid ammonium sulfate was added to the pooled fractions to result in a final concentration of 1.2 M. Insoluble material was pelleted by centrifugation and the complex was applied to a Phenyl Sepharose HP column (CV = 10 mL; Cytiva) equilibrated with 20 mM MOPS-NaOH [pH 6.7 at 4 °C], 1.2 M $(NH_4)_2SO_4$. A linear gradient from 1.2 M to 0 M $(NH_4)_2SO_4$ was used to elute the complex. Fractions containing FimGC were pooled, concentrated using a 10 kDa MWCO Amicon Ultra filter (Merck Millipore) and applied to a Superdex 75 26/60 size-exclusion chromatography column (Cytiva) equilibrated with 20 mM Tris-HCl [pH 8.0 at room temperature], 50 mM NaCl. Appropriate fractions containing FimGC, were pooled after size-exclusion chromatography, flash-frozen in liquid nitrogen and stored at -80 °C before being used in in vitro assembly experiments.

### Purification of FimAC

FimAC was produced and purified as described previously[20,31,51]: FimA was insolubly expressed, purified and then refolded in presence of purified soluble FimC. FimC was produced by growing BL21(DE3) *E. coli* cells carrying plasmid pFimC-cyt in 2YT medium supplemented with 100 µg/mL Ampicillin at 37 °C to an $OD_{600}$ of 1.0, inducing protein production by addition of IPTG to a final concentration of 100 µM and incubating for an additional 4 h at 37 °C. Cells were then harvested by centrifugation (9180 × *g*, 10 min, 4 °C) and resuspended in 100 mM Tris-HCl [pH 8.0 at 4 °C], 1 mM EDTA (3 mL of buffer per g of cell pellet) using a homogenizer. After disrupting the cells using a microfluidizer, the lysate was cleared by centrifugation (48,000 × *g*, 30 min, 4 °C) and the supernatant dialyzed against 10 mM Tris-HCl [pH 8.0 at 4 °C]. The sample was then loaded onto a Q Sepharose FF column (CV = 60 mL; Cytiva) equilibrated with the same buffer and the protein was collected in the flow-through. Following dialysis against 20 mM MES-NaOH [pH 6.0 at 4 °C], FimC was loaded onto a Source 30S column (CV = 17 mL; Cytiva) equilibrated with the same buffer. A linear gradient from 0 to 200 mM NaCl was used to elute the protein. Fractions containing FimC were pooled, and solid ammonium sulfate was added to a final concentration of 1.4 M before loading the sample on a Phenyl Sepharose HP column (CV = 25 mL; Cytiva) equilibrated with 20 mM MES-NaOH [pH 6.0 at 4 °C], 1.4 M $(NH_4)_2SO_4$. A linear gradient from 1.4 M to 0 M $(NH_4)_2SO_4$ was used to elute the protein. Fractions containing FimC were pooled.

For production of FimA, *E. coli* BL21(DE3) carrying plasmid pFimAwt-cyt were grown in 2YT medium supplemented with 100 µg/mL Ampicillin at 37 °C. At an $OD_{600}$ of 1.0, protein expression was induced by addition of IPTG to a final concentration of 100 µM, cells incubated for an additional 4 h at 37 °C and then harvested by centrifugation (9180 × *g*, 10 min, 4 °C). Cells were resuspended and disrupted as described for FimC. To the cell lysate, 0.5 volumes of 60 mM EDTA-NaOH pH 7.0, 1.5 M NaCl, 6% (v/v) Triton X-100 were added, incubated under stirring condition for 30 min at 4 °C. Inclusion bodies were pelleted by

centrifugation (48,000 × *g*, 30 min, 4 °C). The pellet was washed five times with 100 mM Tris-HCl [pH 8.0 at 4 °C], 20 mM EDTA before solubilizing it in 50 mM Tris-HCl [pH 8.0 at 4 °C], 6 M GdmCl, 1 mM EDTA, 50 mM DTT (20 mL buffer per gram of inclusion bodies). Solubilized reduced FimA was applied to a Superdex 75 26/60 size exclusion column equilibrated with 20 mM acetic acid-NaOH [pH 4.0 at room temperature], 6 M GdmCl. Eluted protein was diluted to a final concentration of 5 µM in 6 M GdmCl, 50 mM Tris-HCl [pH 8.0 at 4 °C], 0.1 µM CuCl$_2$ and incubated at room temperature overnight to form the single intramolecular disulfide bond of FimA by copper-catalyzed air oxidation. Oxidized, unfolded protein was concentrated using Hydrosart cassettes (10 kDa MWCO) on a crossflow filtration system (Sartorius) and ultrafiltration. FimAC complex was formed by 30-fold rapid dilution of FimA in 20 mM NaH$_2$PO$_4$-NaOH [pH 7.0 at room temperature], 150 mM NaCl, 1 mM EDTA, EDTA-free protease inhibitor tablets (1 tablet per 50 mL of buffer) in the presence of a 2-fold molar excess of soluble FimC. The complex was applied to a Sephadex G25 desalting column (CV = 530 mL; Cytiva) eluted in 20 mM MES-NaOH [pH 5.5 at 4 °C] and further purified using a Source 30S column equilibrated with the same buffer. A linear gradient from 0 to 200 mM NaCl was used to elute the complex. After concentration by ultrafiltration, the complex was applied to a Superdex 75 26/60 size exclusion column (Cytiva) equilibrated with 20 mM Tris-HCl [pH 8.0 at room temperature], 50 mM NaCl. Appropriate fractions containing FimAC were pooled after size-exclusion chromatography, flash-frozen in liquid nitrogen and stored at −80 °C before being used in in vitro assembly experiments.

### Purification of FimIC$_{His}$

The FimIC$_{His}$ complex was purified similarly to FimAC as described previously[20]. FimC$_{His}$ was produced by growing *E. coli* BL21(DE3) cells carrying plasmid pFimC$_{His}$-cyt in 2YT medium at 37 °C to an OD$_{600}$ of 0.8 and inducing by addition of IPTG to a final concentration of 100 µM. After incubation for an additional 4 h at 37 °C, cells were harvested by centrifugation (9180 × *g*, 10 min, 4 °C), resuspended in 3 mL resuspension buffer (50 mM NaH$_2$PO$_4$-NaOH [pH 8.0 at RT], 300 mM NaCl, 2 mM MgCl$_2$, 1.5 mM PMSF, EDTA-free protease inhibitor tablets) per g of cell pellet, disrupted using a microfluidizer and insoluble material was pelleted by centrifugation (48,000 × *g*, 30 min, 4 °C). The supernatant was loaded onto a HisTrap column (CV = 5 mL, Cytiva) equilibrated with 50 mM NaH$_2$PO$_4$-NaOH [pH 8.0 at RT], 300 mM NaCl, washed with the same buffer containing 20 mM imidazole and then eluted with 250 mM imidazole. Fractions containing FimC$_{His}$ were pooled and dialyzed against 20 mM MOPS-NaOH [pH 6.8 at 4 °C] over night before loading the sample on a self-packed Source 30S column (CV = 14 mL, Cytiva) equilibrated with 20 mM MOPS-NaOH [pH 6.8 at 4 °C]. A linear gradient from 0 to 400 mM NaCl was used to elute the protein. FimC$_{His}$-containing fractions were pooled.

FimI was expressed from plasmid pFimI-cyt and purified after solubilization of inclusion bodies, as described for FimA to yield oxidized, unfolded FimI. To form FimIC$_{His}$ complexes, unfolded, oxidized FimI was rapidly diluted 30-fold in CH$_3$COOH-NaOH [pH 5.0 at 4 °C] containing a 2-fold molar excess of soluble FimC$_{His}$. After 10 minutes of incubation, the complex was applied to a Sephadex G25 column (CV = 530 mL; Cytiva) and eluted in 20 mM MOPS-NaOH [pH 6.7 at 4 °C], followed by cation exchange chromatography on a self-packed Source 30S column (CV = 14 mL; Cytiva). The complex was eluted using a linear gradient from zero to 300 mM NaCl, FimIC$_{His}$-containing fractions were pooled and applied to a Superdex 75 16/60 size exclusion column (Cytiva) equilibrated with 20 mM Tris-HCl [pH 8.0 at room temperature], 50 mM NaCl. Appropriate fractions containing FimIC$_{His}$ were pooled after size-exclusion chromatography, flash-frozen in liquid nitrogen and stored at −80 °C before being used in in vitro assembly experiments.

### Protein concentration determination

Protein concentrations of FimD and all FimC-subunit complexes were determined based on their specific absorbance at 280 nm using the molar extinction coefficients described previously[20].

### Assembly and purification of FimA-bound complex FimDHGFA$_n$C

As a first step of all assembly reactions, FimDCH at a concentration of 0.35 µM was activated by addition of an 8-fold molar excess of FimCG and FimCF over FimD and incubation for 30 min at 23 °C as described previously[20].

In a second step, FimA was added at a five-fold molar excess with respect to FimD and incubated for 60 min at 23 °C. The assembly reaction was performed in 20 mM Tris-HCl [pH 8.0 at RT], 50 mM NaCl, 0.05 % DDM. Subsequently, Amphipol A8-35 was added to the DDM-solubilized protein at an amphipol:FimD mass ratio of 5:1 and the mixture was incubated for four hours. Bio-Beads SM-2 (Bio-Rad Laboratories, Inc.), washed with detergent-free buffer, were then added at a 20-fold mass excess over DDM present in the sample to remove the detergent by incubating at 4 °C overnight. The next day, Bio-Beads were removed by sedimentation. The supernatant was then passed over a gravity flow column packed with Ni-NTA Sepharose 6 FF (CV = 1 mL) equilibrated with 20 mM Tris-HCl [pH 8.0 at RT], 50 mM NaCl. The flow-through was collected and the sample was applied to a Superdex 200 10/300 GL (CV = 24 mL) column (Cytiva) equilibrated with 20 mM Tris-HCl [pH 8.0 at RT], 50 mM NaCl to remove unreacted pilin-chaperone complexes and free FimC. The peak containing the complex of interest was pooled and concentrated to 3.6 mg/mL using an Amicon Ultra 0.5 mL centrifugal filter with a molecular weight cut-off of 100 kDa (Merck Millipore).

### Assembly and purification of the complex FimDHGFA$_n$IC$_{His}$

Initial steps of pilus assembly were performed as described before. FimA was added at a molar ratio of 5:1 with respect to FimD and incubated for 60 min at 23 °C. To terminate pilus assembly, FimI was added at a 10-fold molar excess over FimD and incubated for an additional hour. Amphipol exchange and removal of DDM was performed as described for the FimA-bound complex. The reaction mixture was then passed over a gravity flow column packed with Ni-NTA Sepharose 6 FF (CV = 1 mL) equilibrated with 20 mM Tris-HCl [pH 8.0 at RT], 50 mM NaCl. Bound FimDHGFA$_n$IC$_{His}$ complex was eluted by addition of the same buffer containing 500 mM Imidazole. A 100 kDa MWCO spin filter was used to concentrate the sample and reduce the imidazole concentration by five repetitive steps of concentration to approximately 50 µL followed by dilution with buffer without imidazole to 500 µL. The sample was then concentrated to 3.3 mg/mL.

### Cryo-EM sample preparation

FimDHGFA$_n$C was concentrated to 3.6 mg/mL. Quantifoil R2/2 300 mesh copper grids were glow-discharged with a current of 25 mA for 30 s. Samples (3.5 µL) were applied to the grid and vitrified in a liquid nitrogen-cooled mixture of ethane and propane using a Vitrobot Mark IV (Thermo Fisher Scientific) with the following settings: Blot force: 0; Blot time: 4 s; Wait time 5 s; 100% humidity at 4 °C. FimDHGFA$_n$IC$_{His}$ was concentrated to 3.3 mg/mL. Quantifoil R2/2 300 mesh copper grids were glow-discharged with a current of 25 mA for 45 s. Sample vitrification was performed with the same settings as described earlier.

### Cryo-EM data collection

For the complex FimDHGFA$_n$C, cryo-EM data collection was performed on a Titan Krios G3i microscope (Thermo Fisher Scientific) operated at 300 kV with a Gatan K3 detector and a BioQuantum post-column energy filter with a slit width of 20 eV. Two datasets with 33'005 and 33'828 movies (66,832 movies total) were collected at a nominal magnification of 130,000x (pixel size: 0.65 Å). Data was collected in an automated fashion using EPU software (Thermo Fisher

Scientific) with a target defocus range of −1.0 to −2.8 μm (step size 0.2) and a total electron dose of ~64 e⁻/Å² per exposure (1.1 s, 40 frames).

For the complex FimDHGFA$_n$IC$_{His}$, data was collected on a Titan Krios G4 microscope (Thermo Fisher Scientific) operated at 300 kV and equipped with a Gatan K3 detector and a BioContinuum post-column energy filter set to a slit width of 20 eV. A total of 59,754 movies (of which 6525 movies at a 30° tilt angle) were collected at a nominal magnification of 165,000x (pixel size: 0.51 Å). Data was collected in an automated fashion using EPU (Thermo Fisher Scientific) with a set defocus range of −0.8 to −2.4 μm (step size 0.2) and a total electron dose of ~68 e⁻/Å² per exposure (0.6 s, 40 frames).

## Cryo-EM data processing
A schematic overview of the data processing workflow for complex FimDHGFA$_n$C is shown in Supplementary Figs. 2 and 7, and for complex FimDHGFA$_n$IC$_{His}$ in Supplementary Fig. 12. All movies were subjected to motion correction using PatchMotion in cryoSPARC v4.4.1[52] and CTF estimation was performed using PatchCTF. Exposures with an estimated CTF fit with resolution above 6 Å were excluded.

## Structural analysis of FimDHGFA$_n$C
The data processing workflow for the FimA-bound assembly platform complex FimDHGFA$_n$C is summarized in Supplementary Fig. 2. Approximately 250 particles were picked manually and used to train the neural network-based particle picker Topaz 0.2.4[53] using Conv127 as the model architecture. The generated model was used to pick particles from 500 randomly selected micrographs. Particles were subjected to rounds of 2D classification in cryoSPARC, and the best classes were selected to repeat training of the neural network. From all micrographs, 10,241,804 particles were picked using the improved model, extracted at a box size of 512 pixels and initially Fourier-cropped to a box size of 128 pixels (pixel size: 2.60 Å/px). Different subsets were selected after several rounds of 2D classification and used for ab initio model generation. Particles were subjected to 3D heterogenous refinement with nine classes using multiple ab initio models as references. Two classes with 2,313,645 particles were selected and another round of 3D heterogeneous refinement was performed using the same references as in the previous classification. A single class with 472,076 particles was selected, particles were extracted at a box size of 512 pixels, Fourier-cropped to 256 pixels (pixel size: 1.30 Å/px) and non-uniform refinement yielded a 3D-reconstruction with a resolution of 4.3 Å. The resolution was further improved to 4.0 Å by 3D classification without alignments and a focus mask including only the assembly platform, followed by non-uniform refinement of a single class with 119,055 particles. To account for motion between the pilus rod and the assembly platform, two strategies were used to improve flexible parts of the map: 1) Particles were imported into RELION-5.0[34] using UCSF pyem scripts[54] and further 3D refinement with blush regularization was performed. The 4.1 Å resolution consensus map (map I) was used as input for multi-body refinement[55] with blush regularization[33] and two bodies: body 1 included the pilus rod (3.6 Å, map II), while body 2 included FimD, FimC, FimA$_n$ and FimA$_{n-1}$ (3.9 Å, map III).

2) Local refinement in cryoSPARC[52] was used with either a mask including FimD, FimC, FimA$_n$, and FimA$_{n-1}$ or additionally FimA$_{n-2}$ to yield cryo-EM maps with resolutions of 3.5 Å (map IV) and 3.6 Å (map V), respectively.

## Analysis of the tip fibrillum-to-rod transition using FimDHGFA$_n$C
A single 3D class with 1,019,239 particles showing density for both the pilus rod and the last tip subunit was selected from 3D heterogenous refinement of the initially picked 10,241,804 particles from above (Supplementary Fig. 2). Another round of 3D heterogeneous refinement with three classes was performed which included two references of

just the pilus rod to remove particles which did not show tip fibrillums. Similarly, a single class with 621,220 particles was selected, particles were recentered to have the map center closer to the tip-to-rod transition and extracted at a box size of 512 pixels, Fourier-cropped to 256 pixels (pixel size: 1.30 Å/px). Non-uniform refinement of this subset yielded a cryo-EM map with 2.9 Å resolution. The density of the last tip subunit, FimF, was further improved by 3D classification without alignments and a focus mask around FimF and parts of the previous tip subunit. A single class with 127,989 particles was selected and subjected to a non-uniform refinement resulting in a map with a global resolution of 3.1 Å (map VIII) but improved density for subunit FimF. To resolve different states of tip fibrillum orientation, particles from class 1 and class 3 of the previous 3D classification were extracted at a box size of 800 pixels and Fourier-cropped to 400 pixels (pixel size: 1.30 Å/px). Particles were then imported into RELION-5.0[34] using UCSF pyem scripts[54] and 3D classification without image alignment, with a spherical mask (encompassing the tip subunits incorporated before FimF), and blush regularization[33] was carried out. Classes for which the whole tip fibrillum was resolved were subjected to 3D refinement and post processing in RELION-5.0[34]. Four different distinct orientations of the tip fibrillum could be resolved to 3.9, 4.0, 4.4 and 3.9 Å global resolution (map IX, X, XI, and XII).

## FimDHGFA$_n$C – FimD-tip complexes (conformer 1 and 2)
The data processing workflow for the FimD-tip complexes is summarized in Supplementary Fig. 7. Initially, different subsets of particles were selected after 2D classification and used for ab-initio model generation. Afterwards, the 10,241,804 Topaz-picked particles (see "FimA-bound assembly platform") were subjected to 3D heterogenous refinement with multiple references in CryoSPARC. Two different conformers were selected for further 3D classification:

**conformer 1.** A class of 1,423,150 particles corresponding to conformer 1 of the FimD-tip complex was selected and subjected to 3D heterogenous refinement with 5 classes. A single class of 406,505 particles with well-resolved features was selected and particles were re-extracted at a pixel size of 1.30 Å/px. Non-uniform refinement of these particles yielded a 3D reconstruction at 3.3 Å global resolution. To further improve map quality for the lectin domain of FimH, particles were 3D classified without alignments and the two best of 10 classes with a total of 299,280 particles were selected. After additional clean-up in 2D, non-uniform refinement including defocus and global CTF refinement with the remaining 242,477 particles yielded a structure with a global resolution of 3.3 Å (map VI).

**conformer 2.** A class of 1,211,290 particles corresponding to conformer 2 of the FimD-tip complex was selected and subjected to 3D heterogenous refinement with 4 classes. A single class of 370,262 particles with well-defined features was selected and particles were re-extracted at a pixel size of 1.30 Å/px. Non-uniform refinement yielded a structure with a global resolution of 5.4 Å. 3D classification without alignments with a focus mask excluding the amphipol belt and 5 classes was used to isolate more distinct states of this conformation. The best two, very similar classes with a total of 149,857 particles were selected and further cleaned by 2D classification. Non-uniform refinement with the remaining 98,487 particles and a static mask yielded a final structure with a global resolution of 4.2 Å (map VII).

## Analysis of the FimI-terminated type 1 pilus FimDHGFA$_n$IC$_{His}$
The data processing workflow for the FimI-bound assembly platform complex FimDHGFA$_n$IC$_{His}$ is shown in Supplementary Fig. 12. For particle picking, Topaz 0.2.4[53] was trained on approximately 250 manual particle picks as described previously for the FimDHGFA$_n$C dataset but using model architecture ResNet8. A total of 8,207,346 particles were picked and extracted at a box size of 600 pixels and initially Fourier-

cropped to a box size of 120 pixels (pixel size: 2.55 Å/px). Iterative 2D classifications were used to generate subsets of particles for ab initio models. Multiple ab initio models were used as an input for three consecutive 3D heterogenous refinements in cryoSPARC 4.4.1[52]. In each round, classes with the most well-resolved features were selected and used as an input for the next round of heterogenous refinement. In the third round, a single 3D-class with 222,024 particles was selected and particles were extracted at a box size of 600 pixels and then Fourier-cropped to 300 pixels (pixel size: 1.02 Å/px). Non-uniform refinement yielded a cryo-EM map with a resolution of 4.2 Å. Local refinement with a mask around the assembly platform was used to further improve the resolution to 3.8 Å. The same mask was used for 3D classification without alignment in cryoSPARC[52]. The best of five classes was selected, particles were re-centered to center the map at the transition from the subunit occupying the usher pore to the first subunit exiting the pore. Local refinement gave a map with a resolution of 3.6 Å (map XIII). 3D heterogenous refinement with multiple references derived from 3D classification without alignment in RELION-5.0[34] was then used to separate particles into classes which had I) the pilus rod sitting at an angle on top of the usher, II) multiple subunits incorporated but no pilus rod visible, III) just FimH and two other subunits incorporated, or IV) the first subunit exiting the usher pore sitting at a different angle compared to (I) and the pilus rod sitting on top. Particles of class 1 were then subjected to non-uniform refinement resulting in a map with 6.4 Å resolution (map XIV). A similar multi-body refinement[55] with blush regularization[33] strategy as described for the FimDHGFA$_n$C was then used to improve the resolution of the pilus rod and the assembly platform. Particles were transferred to RELION-5.0[34] using UCSF pyem[54]. Body 1 and its respective mask included the pilus rod and could be refined to 4.0 Å resolution (map XV) while body 2 and its respective mask included the assembly platform and could be refined to 4.4 Å resolution (map XVI). Particles of class 4 which showed a different orientation of the pilus rod when compared to class 1, were subjected to a non-uniform refinement (4.3 Å resolution, map XVIII). The resolution was then further improved by a local refinement with a static mask encompassing the assembly platform and the first subunit exiting the usher pore. This yielded a cryo-EM map with a resolution of 4.2 Å (map XVII).

## Model building and refinement

**FimDHGFA$_n$C – FimA-bound assembly platform.** For initial model building into map II, multiple copies of FimA from PDB-ID 5OH0[10] were placed into the cryo-EM map using rigid-body fitting in UCSF Chimera[56]. The model was then refined by iterative cycles of manual model adjustment in Coot 0.9.8.93[57] followed by *phenix.real_space_refine* in Phenix 1.21-5207-00[58]. In total, nine FimA subunits, the first complemented with a FimA donor strand, were built and refined.

For model III and IV, FimD derived from PDB-ID: 6E15[22], FimCA from PDB-ID: 4DWH[14] and an Alphafold model of FimAt complemented with a FimA donor strand were placed in the cryo-EM map using rigid-body fitting in Chimera. Residues 1–115 of FimD were not built since the NTD was only visible at very low threshold and therefore not well-resolved. The model was subjected to iterative cycles of manual model adjustment in Coot and real space refinement using *phenix.real_space_refine*. For model V, model IV was first rigid-body fitted into the cryo-EM map using Chimera. The model was subjected to iterative cycles of manual model adjustment in Coot and real space refinement using *phenix.real_space_refine*. Before the last iteration of real space refinement, subunit FimA$_{n-2}$ complemented by the Nte of FimA$_{n-1}$ of model II was placed in the cryo-EM map using rigid-body fitting in Chimera and manually connected in Coot.

## FimD-tip fibrillum complex - conformer 1

For building the model of the FimD-tip complex - conformer 1 (map VI), FimD, FimH, FimG, FimF and FimC from PDB-ID 6E15 were manually placed in the cryo-EM density and then rigid-body fitted in UCSF Chimera. The model was first subjected to one round of *phenix.real_space_refine* with rigid-body fitting. This was then followed by iterative cycles of manual model adjustment in Coot and real space refinement using *phenix.real_space_refine*.

## FimDHGFA$_n$C – tip-to-rod transition

To build model VIII, multiple copies of FimA from PDB-ID 5OH0[10] and FimF of PDB-ID: 3JWN[23] were placed in cryo-EM map VIII using rigid-body fitting in UCSF Chimera. This was followed by iterative cycles of manual adjustment in Coot and *phenix.real_space_refine*.

## FimDHGFA$_n$IC$_{His}$ – FimI-bound assembly platform

For model XIII, FimI and FimC from PDB-ID 6SWH and FimD from model IV were placed in the cryo-EM density using rigid-body fitting in UCSF Chimera. The model was then subjected to iterative cycles of manual model adjustment in Coot and real space refinement using *phenix.real_space_refine*.

For model XV, multiple copies of FimA from PDB-ID 5OH0[10] were rigid-body fitted into the cryo-EM map using UCSF Chimera. The model was then subjected to iterative cycles of manual model adjustment in Coot and real space refinement using *phenix.real_space_refine*.

To build model XVI, FimD, FimI and FimC from model XIII as well as FimA from model IV were placed in the cryo-EM density using rigid-body fitting in UCSF Chimera. The model was then subjected to iterative cycles of manual model adjustment in Coot and real space refinement using *phenix.real_space_refine*.

All real space refinements in Phenix[58] were performed using secondary structure and geometric restraints. MolProbity[59] was used for model validation.

## Analysis of the angle between the pilus rod and outer membrane

Membrane boundaries were calculated using the PPM 3.0 web server[60] using the model built into body 2 of either the FimA- or FimI-bound complex (model III or XVI, respectively) as an input. The output was rigid-body fitted into body 2 of each multi body refinement. Corresponding models built into body 1 (model IV or XV) were also rigid-body fitted. The plane of the outer membrane boundary was calculated using the *define plane* command in ChimeraX. The *define centroid* command was used to get the centroid of the three FimA subunits at the bottom of the fitted pilus rod structure and a second centroid was defined based on the three uppermost FimA subunits. The axis through the two centroids was calculated with the *define axis* command. The angle between the pilus rod axis and the plane was calculated using the *angle* command.

## Analysis of rod flexibility

The model built into body 1 (pilus rod) was fitted into the first and last bin (angular distribution maxima) of each of the first three components of multi body refinement using rigid-body fitting in Chimera to determine the angular difference between the maxima of pilus rod movement. Fitted models were saved relative to their respective map. Maps and models were opened in ChimeraX. The *define centroid* command was used to get the centroid of the three FimA subunits at the bottom of the fitted pilus rod structure and a second centroid was defined based on the three uppermost FimA subunits. The axis through the two centroids was then calculated with the *define axis* command. After defining the central axis for the two boundaries, the angle between the two axes was determined with the *angle* command.

## Analysis of tip fibrillum flexibility

After 3D classification, classes with distinct angles of the tip were selected. FimF-FimA$_1$ of model VIII was placed in each map and rigid-body fitted in ChimeraX using the *fitmap* command. Similarly, FimH-FimG derived from PDB ID 3JWN was fitted into the tip density. The

central axes of FimF and FimG were determined using the *define axis* command in ChimeraX and the angle between the axes was then calculated using the *angle* command.

## Multiple sequence alignment

Protein sequences of FimF, FimA and tip adaptor subunits SfaG, FmlF, PapK and YfcP were aligned using ClustalΩ[61] as part of the MPI Bioinformatics Toolkit[62,63].

## Prediction of Chaperone-Usher pilus tip-to-rod transitions using Alphafold

Models of tip-to-rod transitions were generated using AlphaFold2-multimer[37] implemented in ColabFold 1.5.2[64] and installed locally as LocalColabFold (https://github.com/YoshitakaMo/localcolabfold) using default settings and Amber relaxation. Predictions were split by first generating a model including two tip subunits and one rod subunit complemented with the donor strand of another rod subunit. This was followed by a second prediction with four pilus rod subunits to generate a second model of a full turn of the rod. A composite model was then created by aligning both models based on the rod subunit of the first model and the first subunit of the rod from the second model in UCSF ChimeraX.

A model of the F1C/S tip-to-rod transition (SfaS:SfaG:SfaA:SfaA$_{(1-18)}$; pLDDT = 95.2; pTM = 0.835) was generated with four recycles. An F1C/S rod model (SfaA$_{(1-18)}$:SfaA:SfaA:SfaA:SfaA; pLDDT = 94.2; pTM = 0.84) was generated with 11 recycles. A model of the F9 pilus tip-to-rod transition (FmlG:FmlF:FmlA:FmlA$_{(1-26)}$; pLDDT = 92.9; pTM = 0.81) was generated with five recycles. An F9 pilus rod model (FmlA$_{(1-26)}$:FmlA:FmlA:FmlA:FmlA; pLDDT = 89.9; pTM = 0.811) was generated with two recycles. P pilus tip-to-rod transition model (PapE:PapK:PapA:PapA$_{(1-21)}$; pLDDT = 93.8; pTM = 0.865) was generated with three recycles. This model was aligned to the cryo-EM structure of the P pilus (PDB ID 5FLU) based on PapA.

A model of the Yfc pilus tip-to-rod transition (YfcQ:YfcP:YfcV:YfcV$_{(1-17)}$; pLDDT = 93.9; pTM = 0.828) was generated with five recycles. The Yfc pilus rod model (YfcV$_{(1-17)}$:YfcV:YfcV:YfcV:YfcV; pLDDT = 94.5; pTM = 0.89) was generated with six recycles.

## Structural comparison

Interface areas between Fim subunits were calculated using the PISA webserver[65]. Structural alignment of transmembrane and plug domains of FimD was performed using the GESAMT algorithm[66] as part of the CCP4 software suite[67]. The transformation matrix outputted by GESAMT was used to align the models in PyMOL[68].

## Figure preparation

Figures of Cryo-EM maps and corresponding models were prepared using PyMOL[68], UCSF Chimera[56], and UCSF ChimeraX[69]. FSC curves were plotted using OriginPro 2022b (OriginLab). Multiple sequence alignments were visualized with Jalview[70]. Adobe Illustrator was used to compose Figures.

## Reporting summary

Further information on research design is available in the Nature Portfolio Reporting Summary linked to this article.

## Data availability

The cryo-EM maps generated in this study have been deposited in the Electron Microscopy Data Bank (EMDB) under accession codes EMD-50755 (map I – Pilus rod and FimA-bound usher), EMD-50751 (map II – FimDHGFA$_n$C Pilus rod – Multibody 1), EMD-50828 (map III – FimDHGFA$_n$C Usher – Multibody 2), EMD-50829 (map IV – FimDHGFA$_n$C Local refinement 1), EMD-50838 (map V – FimDHGFA$_n$C Local refinement 2), EMD-50861 (map VI – FimD-tip conformer 1), EMD-50954 (map VII – FimD-tip conformer 2), EMD-50839 (map VIII –

Tip-to-rod-transition), EMD-50773 (map IX – Pilus tip and rod – Conformer 1), EMD-50796 (map X – Pilus tip and rod –. Conformer 2), EMD-50806 (map XI – Pilus tip and rod –. Conformer 3), EMD-50809 (map XII – Pilus tip and rod –. Conformer 4), EMD-50843 (map XIII – FimI-bound usher), EMD-50810 (map XIV – Pilus rod and FimI-Bound usher), EMD-50846 (map XV – FimDHGFA$_n$IC$_{His}$ Pilus rod - Multibody 1), EMD-50847 (map XVI – FimDHGFA$_n$IC$_{His}$. Usher - Multibody 2), EMD-50853 (map XVII – FimDHGFA$_n$I$_2$C$_{His}$ Local refinement), and EMD-50812 (map XVIII – FimDHGFA$_n$I$_2$C$_{His}$ Pilus rod and usher). Model coordinates generated in this study have been deposited in the Protein Data Bank (PDB) under accession codes 9FTT (model II – FimDHGFA$_n$C Pilus rod – Multibody 1), 9FW9 (model III – FimDHGFA$_n$C Usher – Multibody 2), 9FWB (model IV – FimDHGFA$_n$C Local refinement 1), 9FWZ (model V – FimDHGFA$_n$C Local refinement 2), 9FY9 (model VI – FimD-tip conformer 1), 9FX0 (model VIII – Tip-to-rod-transition), 9FX8 (model XIII – FimI-bound usher), 9FXA (model XV – FimDHGFA$_n$IC$_{His}$ Pilus rod - Multibody 1), 9FXB (model XVI – FimDHGFA$_n$IC$_{His}$. Usher - Multibody 2), 9FXS (model XVII – FimDHGFA$_n$I$_2$C$_{His}$ Local refinement). Previously published model coordinates used for structural comparison and model building are available in the PDB under accession codes 6E14, 6E15, 6SWH, 5OH0, 4DWH, 3JWN and 5FLU. All plasmids for recombinant protein production used in this study are available upon request.

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

## Acknowledgements

We thank E. Weber-Ban and I. Meuskens for revision of this manuscript. We thank C. Giese for helpful discussions and comments. We thank M. Peterek and B. Qureshi at the Scientific Center for Optical and Electron Microscopy (ScopeM) for their technical support with cryo-EM data acquisition. This work was supported by the Swiss National Science Foundation (Grant 310030_201234).

## Author contributions

P.B. and R.G. designed experiments. P.B. purified proteins, performed and optimized in vitro assembly reactions, prepared cryo-EM grids, collected cryo-EM datasets for all in vitro-assembled complexes and performed model building and refinement. P.A. assisted with screening of cryo-EM grids and data collection. P.B., with the support of P.A. and D.B., performed single-particle cryo-EM data processing and analysis. R.G. supervised the work and acquired funding. P.B. prepared the figures and wrote the initial draft of the manuscript. All authors contributed to the writing of the final manuscript.

## Competing interests

The authors declare no competing interests.
