## [Peer Review file · Nature Communications]

Structures of the Escherichia coli Type 1 Pilus during Pilus Rod Assembly and after Assembly Termination

Corresponding Author: Professor Rudi Glockshuber

Version 0:

Reviewer comments:

Reviewer #1

(Remarks to the Author)

Bachmann et al. have solved the cryo-EM structures of a reconstructed type 1 pilus in complex with the secretion platform (usher), representing the entire organelle in both the pilus rod assembly state and the completed, terminated assembly state. Previously, similar structures were determined by X-ray crystallography to capture the assembly process of the pilus fibrillum. These new structures add important structural knowledge on the processes of assembly and termination of the pilus rod. They also demonstrate how the adaptor subunit breaks the helical symmetry of the pilus rod to position the fibrillum. The authors employed recent advances in single-particle cryo-EM to solve the structures, which worked beautifully. The paper is well-written and well-illustrated. I have only a few comments:

1. While the new structures are novel and important, they mostly confirm previous conclusions obtained from crystal structures rather than providing completely new insights into the process. In my opinion, the title "Structural Basis of Type 1 Pilus Assembly and Termination in Uropathogenic E. coli" is overstated. Additionally, there have been papers with similar titles, such as "Molecular Mechanism of P Pilus Termination in Uropathogenic Escherichia coli." Could the authors make the title more specific? Similarly, in the abstract, the authors stated, "However, the structural basis of type 1 pilus assembly and termination is unknown." Could the authors revise this to something like, "However, the precise mechanism of type 1 pilus assembly and termination on the usher is unknown"?
2. Pages 13-14: "However, when comparing the FimA- and FimI-bound usher structures to the FimF-bound structure (which has an open P5 pocket), we found that the P5 pocket also remained closed when FimI was absent (Supplementary Figure 19) [22]." Do the authors mean that the P5 pocket in FimA remained closed during assembly? Please clarify this sentence.
3. Page 14, end of the Results: In the Yersinia pestis Caf (F1 capsule) system, the P5 pocket is occupied by the chaperone, yet assembly is very efficient (Yu et al. JMB 2012). It's possible that in FimA, the P5 pocket is only partially or temporarily closed, which does not affect assembly, while it's fully closed in FimI, and the acceptor cleft in FimI is not compatible with the donor strand of FimA, making termination permanent.

Reviewer #2

(Remarks to the Author)

The Bachmann et al. manuscript reports high quality cryoEM structures of the uropathogenic E. coli type 1 pilus complexes in the process of pilus assembly (FimDHGFAnC; usher, fibrillar pilus tip, rod, chaperone) and post-assembly (FimDHGFAnCHis; usher, fibrillar pilus, assembly terminator, chaperone). The complexes were obtained by in vitro assembly in detergent whereby individual components were precisely titrated to obtain complexes of workable length for cryoEM structure determination, and at different stages of assembly. These structures provide key insights into the mechanism of chaperone-usher pilus assembly, including the nature of the transition between the extended fibrillar tip and the coiled helical rod, the incorporation of FimA into the rod, and the mechanism of FimI-mediated assembly termination. The structures are quite remarkable and provide a detailed view of the assembly process. The manuscript is well-written and rigorous and the figures are informative. Below are comments and suggestions, mostly editorial, ordered as issues appear in the manuscript

- Title (Structural Basis of Type 1 Pilus Assembly and Termination in Uropathogenic E. coli) and Abstract statement (“However, the structural basis of type 1 pilus assembly and termination is unknown.”) - Both seem overstated given the plethora of structural and biochemical data published previously by this group and others on chaperone usher pilus assembly. From the detailed structural analysis the authors propose several explanations for key steps in assembly (eg. how FimF directs coiling of FimA in the rod, interactions that stabilize the FimI:FimC complex to terminate pilus assembly) and provide sequence conservation data to support these claims but do not introduce mutations to test these models. Thus, while the manuscript builds on our understanding of chaperone-usher pilus assembly in a number of important ways it does not in itself provide the structural basis for assembly and termination. I recommend the title and abstract statement be modified accordingly.
- A number of figures (eg. Fig. 1e, 2b, 3d) show side chain conformations, atomic positions and interactions that likely were not revealed in the 3.5-6-Angstrom cryoEM maps. The authors should either show maps for these regions to justify the models or comment on the confidence of atom positions in light of the limited resolution.
- It is difficult to distinguish subunits in some figures (eg. Fig. 1d-f, Fig. 2b, 4a) due to the subtle differences in their blue shades. Choose higher contrast colors or provide more labelling to distinguish subunits. Additionally, some of the labels are hard to read as they are grey on grey or light blue on grey (eg. Fig. 4).
- Paragraph starting at Line 149 could be clearer. Line 149 – For clarity, revise sentence: “The overall domain arrangement of the FimDHGFAnC complex is similar ...”. The paragraph would be easier to read if the subunit number (n, n-1, n-2, n-3) were indicated throughout, as well as in each image of Fig. 1d-f.
- Paragraph starting at Line 203. Line 211-213: “Importantly, the chemical identity of Lys35 and Gln36 is conserved across FimF homologs of other chaperone-usher pili that break the helical rod symmetry at the transition to the tip fibrillum (Supplementary Figure 10b).” Since the tip fibrillum forms before the rod is it correct to say that FimF breaks the helical rod symmetry? Isn't it more accurate to say that FimF does fit into the helically symmetrical coil formed by FimA subunits? The authors state later, in the paragraph starting at Line 245, that FimF appears to facilitate the formation of the first turn of the rod by interacting with FimA1, 2 and 3. So instead of “breaking the helical rod symmetry” FimF appears to initiate helical symmetry for FimA. The authors could better describe here how the FimF:FimA interaction differs from the FimA:FimA interaction, preventing FimF from adopting a FimA-like position – presumably it is because FimF interacts with FimA1/FimA3 in a way that is incompatible with a coiled/helical arrangement.
- Fig. 2. It is difficult to relate the left inset in Fig. 2b with the zoomed out picture as they are in different orientations. Provide more labels for both, including for the FimA2 donor strand, which is indistinguishable in color from FimA1. Also label the β C- β D loop in the right inset.
- Line 243-244 – “This underlines the critical role of FimF homologs that function as adaptors aligning the tip fibrillum with the axis of the pilus rod, which also ensures maximum distance of the tip adhesin (FimH in the case of type 1 pili) from the outer bacterial membrane.” It is not clear that the FimF orientation is critical for maximizing this distance – the number of FimA subunits in the rod likely control the pilus length. More likely the extended conformation of the FimF adaptor and of the rest of the fibrillum functions to increase accessibility of the tip to buried N-glycans on host cells.
- Fig. 3c and Sup. Fig. 18. In these figures it is difficult to resolve FimA, FimC and FimI. Show superpositions without FimD.
- Fig. 4. As for Fig. 2b, it is difficult to relate the insets with the zoomed out images as they are in different orientations and the β -strands are not labeled consistently in each. And in panel a it is hard to distinguish FimA from FimC as their colors are so similar.
- Line 268/269 – What is the “FimA-bound sample”? The sample lacking FimI? Clarify.
- Line 325/326: “In contrast to FimA and FimF, FimI does not only form hydrogen bonds with the neighboring A1' strand of FimC.” Sentence is awkward and should be reworded.
- The authors propose that the additional hydrogen bonds observed between FimI and FimC stabilize this complex and prevent dissociation to terminate pilus assembly. Would it be worth testing the role of FimI Arg25 by changing this to a smaller or uncharged residue and showing that it is less efficient at terminating? Or is it only one of several interactions that increase the stability of the FimI:FimC complex relative to that of FimA:FimC?

Minor edits:

Line 59 – can you really say that the FimA interactions in the filament are “infinitely” stable?

Line 87 – add “of FimD” to end of sentence

Line 167 – change “could not be” to “was not”

Line 203 – Change “Specific” to “Specifically”

Version 1:

Reviewer comments:

Reviewer #1

(Remarks to the Author)

The authors have fully addressed my comments. In my opinion, the manuscript is ready for publication. It represents an important contribution to our understanding of pilus assembly via the chaperone-ushe pathway.

Reviewer #2

(Remarks to the Author)

The authors have addressed my concerns satisfactorily.

Point-by point response to the comments of the reviewers:

Blue: Reviewer comments; Black: Our response

Changes in the revised manuscript relative to first version are highlighted with yellow background in the text.

Reviewer #1 (Remarks to the Author):

Bachmann et al. have solved the cryo-EM structures of a reconstructed type 1 pilus in complex with the secretion platform (usher), representing the entire organelle in both the pilus rod assembly state and the completed, terminated assembly state. Previously, similar structures were determined by X-ray crystallography to capture the assembly process of the pilus fibrillum. These new structures add important structural knowledge on the processes of assembly and termination of the pilus rod. They also demonstrate how the adaptor subunit breaks the helical symmetry of the pilus rod to position the fibrillum. The authors employed recent advances in single-particle cryo-EM to solve the structures, which worked beautifully. The paper is well-written and well-illustrated. I have only a few comments:

1. While the new structures are novel and important, they mostly confirm previous conclusions obtained from crystal structures rather than providing completely new insights into the process. In my opinion, the title “Structural Basis of Type 1 Pilus Assembly and Termination in Uropathogenic *E. coli*” is overstated. Additionally, there have been papers with similar titles, such as “Molecular Mechanism of P Pilus Termination in Uropathogenic *Escherichia coli*.” Could the authors make the title more specific? Similarly, in the abstract, the authors stated, “However, the structural basis of type 1 pilus assembly and termination is unknown.” Could the authors revise this to something like, “However, the precise mechanism of type 1 pilus assembly and termination on the usher is unknown”?

Response: We thank the reviewer for the comment. We have changed the title of the manuscript to “Structures of the *Escherichia coli* Type 1 Pilus during Pilus Rod Assembly and After Assembly Termination.” We also followed the reviewer’s suggestion and revised the sentence in the abstract accordingly.

2. Pages 13-14: “However, when comparing the FimA- and FimI-bound usher structures to the FimF-bound structure (which has an open P5 pocket), we found that the P5 pocket also remained closed when FimI was absent (Supplementary Figure 19) [22].” Do the authors mean that the P5 pocket in FimA remained closed during assembly? Please clarify this sentence.

Response: We formulated the sentence more precisely as follows (page 14, lines 356-359): “However, when comparing the usher complexes with FimA or FimI as the last incorporated subunit with the usher-tip fibrillum complex (last subunit: FimF, with open P5 pocket), we found that the P5 pocket was closed in the terminal FimA subunit as well as in FimI (**Error! Reference source not found.**) [22].“

3. Page 14, end of the Results: In the *Yersinia pestis* Caf (F1 capsule) system, the P5 pocket is occupied by the chaperone, yet assembly is very efficient (Yu et al. JMB 2012). It’s possible that in FimA, the P5 pocket is only partially or temporarily closed, which does not affect

assembly, while it's fully closed in FimI, and the acceptor cleft in FimI is not compatible with the donor strand of FimA, making termination permanent.

Response: We thank the reviewer for this excellent comment and agree that the role of an accessible P5 pocket in subunit assembly kinetics may have been overestimated. At the end of the *Results* part, we now mention that an occupied P5 pocket in Caf1/Caf1M complexes of the related Caf1 (F1 capsule) system of *Yersinia pestis* does not prevent efficient assembly of Caf1 subunits as observed by Yu et al. (page 14, 361-364) We also would like to thank the reviewer for pointing at a potential incompatibility between the acceptor cleft of FimI and the donor strand of FimA. In the revised manuscript, we now mention this possibility on page 14 (lines 361-362).

Reviewer #2 (Remarks to the Author):

The Bachmann et al. manuscript reports high quality cryoEM structures of the uropathogenic *E. coli* type 1 pilus complexes in the process of pilus assembly (FimDHGFAnC; usher, fibrillar pilus tip, rod, chaperone) and post-assembly (FimDHGFAnICHis; usher, fibrillar pilus, assembly terminator, chaperone). The complexes were obtained by in vitro assembly in detergent whereby individual components were precisely titrated to obtain complexes of workable length for cryoEM structure determination, and at different stages of assembly. These structures provide key insights into the mechanism of chaperone-usher pilus assembly, including the nature of the transition between the extended fibrillar tip and the coiled helical rod, the incorporation of FimA into the rod, and the mechanism of FimI-mediated assembly termination. The structures are quite remarkable and provide a detailed view of the assembly process. The manuscript is well-written and rigorous and the figures are informative. Below are comments and suggestions, mostly editorial, ordered as issues appear in the manuscript

- Title (Structural Basis of Type 1 Pilus Assembly and Termination in Uropathogenic *E. coli*) and Abstract statement (“However, the structural basis of type 1 pilus assembly and termination is unknown.”) - Both seem overstated given the plethora of structural and biochemical data published previously by this group and others on chaperone usher pilus assembly. From the detailed structural analysis the authors propose several explanations for key steps in assembly (eg. how FimF directs coiling of FimA in the rod, interactions that stabilize the FimI:FimC complex to terminate pilus assembly) and provide sequence conservation data to support these claims but do not introduce mutations to test these models. Thus, while the manuscript builds on our understanding of chaperone-usher pilus assembly in a number of important ways it does not in itself provide the structural basis for assembly and termination. I recommend the title and abstract statement be modified accordingly.

Response: Reviewer 1 addressed the same point. We agree and changed the title of the manuscript to “Structures of the *Escherichia coli* Type 1 Pilus during Pilus Rod Assembly and After Assembly Termination”, which addresses our results more specifically.

- A number of figures (eg. Fig. 1e, 2b, 3d) show side chain conformations, atomic positions and interactions that likely were not revealed in the 3.5-6-Angstrom cryoEM maps. The authors should either show maps for these regions to justify the models or comment on the confidence of atom positions in light of the limited resolution.

Response: We agree that the resolutions of our structures in certain areas of the maps are not sufficient for a precise description of side chain conformations and atom positions. At the same time, the local resolution in other areas is significantly higher than the globally stated resolutions of 3.1 to 4.3 Å and allows us to speculate about potential hydrogen bond formation. For Figs. 1d-f, Fig.2b and 3d, we refer to this fact by using the term “potential hydrogen bonds” in the corresponding figure legend. For Figure 2b depicting the interfaces between FimF and the neighboring FimA subunits, the corresponding map is shown in Suppl. Fig. 9. We also added the relevant regions shown in the other main figures to the selection of example densities shown in Suppl. Fig. 4 and Suppl. Fig. 14.

- It is difficult to distinguish subunits in some figures (eg. Fig. 1d-f, Fig. 2b, 4a) due to the subtle differences in their blue shades. Choose higher contrast colors or provide more labelling to distinguish subunits. Additionally, some of the labels are hard to read as they are grey on grey or light blue on grey (eg. Fig. 4).

Response: We thank the reviewer for this comment. The color of subunit FimA_{n-1} was changed in Figure 1d-f to increase the contrast relative to the adjacent subunits. Similarly, the colors of subunits were changed in Figure 2a+b as well as in Fig.4 to increase the contrast, making the different subunits easier to distinguish. In addition, we changed the colors of the labels to darker colors to improve readability.

- Paragraph starting at Line 149 could be clearer. Line 149 – For clarity, revise sentence: “The overall domain arrangement of the FimDHGFAnC complex is similar ...”. The paragraph would be easier to read if the subunit number (n, n-1, n-2, n-3) were indicated throughout, as well as in each image of Fig. 1d-f.

Response:

We now formulated the first sentence of the paragraph more precisely and also provide the specific numbering of the FimA subunits in the paragraph (page 7, lines 150-154). Accordingly, we added specific labels to each FimA subunit in Fig. 1d-f.

- Paragraph starting at Line 203. Line 211-213: “Importantly, the chemical identity of Lys35 and Gln36 is conserved across FimF homologs of other chaperone-usher pili that break the helical rod symmetry at the transition to the tip fibrillum (Supplementary Figure 10b).” Since the tip fibrillum forms before the rod is it correct to say that FimF breaks the helical rod symmetry? Isn’t it more accurate to say that FimF does fit into the helically symmetrical coil formed by FimA subunits? The authors state later, in the paragraph starting at Line 245, that FimF appears to facilitate the formation of the first turn of the rod by interacting with FimA1, 2 and 3. So instead of “breaking the helical rod symmetry” FimF appears to initiate helical symmetry for FimA. The authors could better describe here how the FimF:FimA interaction differs from the FimA:FimA interaction, preventing FimF from adopting a FimA-like position – presumably it is because FimF interacts with FimA1/FimA3 in a way that is incompatible with a coiled/helical arrangement.

Response: We fully agree with the reviewer that “breaking the helical rod symmetry” was a potentially misleading description for FimF at the transition between the tip fibrillum and the pilus rod. We thus removed “breaking the helical rod symmetry” from the text (page 8, lines

213-215). Additionally, we added a sentence describing potential mechanistic implications, which the differences in the FimF-FimA vs FimA-FimA interface might have on the positioning of FimF (page 9, lines 221-224).

- Fig. 2. It is difficult to relate the left inset in Fig. 2b with the zoomed out picture as they are in different orientations. Provide more labels for both, including for the FimA2 donor strand, which is indistinguishable in color from FimA1. Also label the β C- β D loop in the right inset.

Response: We revised Fig. 2b according to all suggestions of the reviewer. In addition, we added an arrow indicating the rotation of the left inset relative to the structure in the center of Fig. 2b.

- Line 243-244 – “This underlines the critical role of FimF homologs that function as adaptors aligning the tip fibrillum with the axis of the pilus rod, which also ensures maximum distance of the tip adhesin (FimH in the case of type 1 pili) from the outer bacterial membrane.” It is not clear that the FimF orientation is critical for maximizing this distance – the number of FimA subunits in the rod likely control the pilus length. More likely the extended conformation of the FimF adaptor and of the rest of the fibrillum functions to increase accessibility of the tip to buried N-glycans on host cells.

Response: We agree with the reviewer and removed the statement about the role of FimF on the distance between FimH and the outer membrane. We also included a new sentence addressing a possible function of FimF for increasing the accessibility of FimH at the distal end of the tip fibrillum for buried glycan receptors on host cells. The paragraph now reads as follows (page 10, lines 245-248):

“This underlines the critical role of FimF homologs that function as adaptors aligning the tip fibrillum with the axis of the pilus rod. The extended conformation of the FimF adaptor and the rest of the tip fibrillum may also function to increase the accessibility of FimH to buried high-mannose type N-glycans on host cells.”

- Fig. 3c and Sup. Fig. 18. In these figures it is difficult to resolve FimA, FimC and FimI. Show superpositions without FimD.

Response: FimD was now made transparent in Fig. 3c to ensure that FimA, FimC and FimI are easily visible. The domain labels were removed to put the focus on the positioning of the subunits occupying the pore of the usher and the last, FimC-capped subunit on the periplasmic side. Supplementary Figure 18 shows how the angle between FimA and FimI observed in the crystal structure of the FimAIC_{His} complex would result in a clash with the transmembrane domain of FimD in the structure of FimDHGFA_nIC. We therefore kept the superposition as it was.

- Fig. 4. As for Fig. 2b, it is difficult to relate the insets with the zoomed out images as they are in different orientations and the β -strands are not labeled consistently in each. And in panel a it is hard to distinguish FimA from FimC as their colors are so similar.

Response: We now indicate how all three insets in Fig. 4 a-c were rotated relative to the corresponding cartoon representations at the top. The different orientation of the insets was chosen to assure visibility of strands β B and β A' of the subunits FimA_n, FimI and FimF. In addition, we improved the labels of the β -strands, and a darker color was used for better distinction between FimA and FimC.

- Line 268/269 – What is the “FimA-bound sample”? The sample lacking FimI? Clarify.

Response: “FimA-bound sample” was replaced by “FimDHGFA_nC sample” (page 10, line 272-273)

- Line 325/326: “In contrast to FimA and FimF, FimI does not only form hydrogen bonds with the neighboring A1’ strand of FimC.” Sentence is awkward and should be reworded.

Response: The sentence was reworded as follows: “In contrast to FimA and FimF, FimI does not only form hydrogen bonds with the neighboring A1’ strand of FimC, but also with the B1 strand of FimC” (page 13, lines 330-332)

- The authors propose that the additional hydrogen bonds observed between FimI and FimC stabilize this complex and prevent dissociation to terminate pilus assembly. Would it be worth testing the role of FimI Arg25 by changing this to a smaller or uncharged residue and showing that it is less efficient at terminating? Or is it only one of several interactions that increase the stability of the FimI:FimC complex relative to that of FimA:FimC?

Response: There are indeed several additional interactions between FimI and FimC that appear to stabilize the FimI-FimC relative to the FimA-FimC complex. For example, Glu27 of FimI likely forms hydrogen bonds with both neighboring A1’ and B1 strands of FimC in a similar fashion as Arg25. Thus, the increased stability against dissociation is likely a combined effect of all additional hydrogen bonds. A comprehensive analysis of the effects of point mutations in FimI would require a systematic analysis of FimD-catalyzed assembly kinetics *in vitro* and of the resulting pilus lengths distributions using FimI variants instead of FimI wild type. This approach would be analogous to our previous publication on stochastic chain termination by FimI (Giese et al. (2023), Nat. Comm. 14, p. 7718) and is beyond the scope of this study.

Minor edits:

Line 59 – can you really say that the FimA interactions in the filament are “infinitely” stable? We kept “infinitely stable against dissociation”, because the extrapolated half-life of spontaneous dissociation of FimA subunits in the pilus rod in the absence of denaturant is longer than the age of the universe (10^{35} years, see reference 31).

Line 87 – add “of FimD” to end of sentence

We added “of FimD” at the end of the sentence (page 4, line 88)

Line 167 – change “could not be” to “was not”

The text was corrected as suggested, (page 7, line 169)

Line 203 – Change “Specific” to “Specifically”

“Specific” was changed to “Specifically”. (page 8, line 205)